# Sex and *APOE* ε4 genotype modify the Alzheimer's disease serum metabolome

Matthias Arnold ⓘ et al.#

Late-onset Alzheimer's disease (AD) can, in part, be considered a metabolic disease. Besides age, female sex and *APOE* ε4 genotype represent strong risk factors for AD that also give rise to large metabolic differences. We systematically investigated group-specific metabolic alterations by conducting stratified association analyses of 139 serum metabolites in 1,517 individuals from the AD Neuroimaging Initiative with AD biomarkers. We observed substantial sex differences in effects of 15 metabolites with partially overlapping differences for *APOE* ε4 status groups. Several group-specific metabolic alterations were not observed in unstratified analyses using sex and *APOE* ε4 as covariates. Combined stratification revealed further subgroup-specific metabolic effects limited to *APOE* ε4+ females. The observed metabolic alterations suggest that females experience greater impairment of mitochondrial energy production than males. Dissecting metabolic heterogeneity in AD pathogenesis can therefore enable grading the biomedical relevance for specific pathways within specific subgroups, guiding the way to personalized medicine.

Female sex is regarded a major risk factor for Alzheimer's disease (AD). Of 5.3 million people in the United States diagnosed with AD at age 65 or older, >60% are women. The lifetime risk of developing AD at age 45 may be almost double in females than in males[1,2], though the exact role and magnitude of a sexual dimorphism in predisposition and progression to AD are controversial[3–6]. Although age is the strongest risk factor for late-onset AD (LOAD), the higher life expectancy of women only partially explains the observed sex difference in frequency and lifetime risk[7]. Complexity is added by studies showing a significant sex difference in effects of the *APOE* ε4 genotype, the strongest common genetic risk factor for LOAD. These studies report risk estimates for ε4 carriers being higher in females, a finding that seems to be additionally dependent on age[8–13]. *APOE* ε4 is also associated with AD biomarkers in a sex-dependent way with larger risk estimates for women than for men[9,14–17], although these findings have not been fully consistent across studies[16,18]. In addition, studies suggest that sex differences in AD may change during the trajectory of disease[19]. Overall risk for mild cognitive impairment (MCI), the prodromal stage of AD, is higher in males[20,21], whereas progression to AD occurs faster in females, at least partly in *APOE* ε4-dependent ways[3,8,10,19,22,23]. The mechanisms underlying this sex-linked and partly intertwined age- and *APOE* ε4-dependent heterogeneity in AD susceptibility and severity are only beginning to unravel, calling for novel approaches to further elucidate molecular sex differences in AD risk and biomarker profiles.

Interestingly, all three of the aforementioned major AD risk factors, age, *APOE* ε4 genotype, and sex, have a profound impact on metabolism[24–30], supporting the view of AD as a metabolic disease[31–33]. In recent years, availability of high-throughput metabolomics techniques, which can measure hundreds of small biochemical molecules (metabolites) simultaneously, enabled the study of metabolic imprints of age, genetic variation, and sex very broadly, covering the entire metabolism: (i) Age-dependent differences were observed in levels of phosphatidylcholines (PCs), sphingomyelins (SMs), acylcarnitines, ceramides, and amino acids[29,34]. A panel of 22 independent metabolites explained 59% of the total variance in chronological age in a large twin population cohort. In addition, one of these metabolites, C-glycosyl-tryptophan, was associated with age-related traits including bone mineral density, lung[30], and kidney function[35]. (ii) As expected from APOE's role in cholesterol and lipid metabolism[36,37], common genetic variants in *APOE* were associated with blood cholesterol levels in genome- and metabolome-wide association studies[37,38]. In addition, associations with levels of various SMs were identified[39,40]. (iii) Analogous to age, sex also affects blood levels of many metabolites from a broad range of biochemical pathways. In a healthy elderly population with mostly postmenopausal women, females showed higher levels of most lipids except lyso-PCs. Levels of most amino acids including branched chain amino acids (BCAAs) were higher in males, though glycine and serine levels were higher in women[24,25]. Gonzalez-Covarrubias et al.[29] reported sex-specific lipid signatures associated with longevity in the Leiden Longevity Study. In women, higher levels of ether-PC and SM species were associated with longevity; no significant differences were observed in men. Thus, based on results from large-scale metabolomics studies, aging may influence a wider range of metabolites in women than men, highlighting the need for sex-stratified analyses.

Many metabolites affected by female sex, age, and *APOE* genotype (e.g., BCAAs, glutamate, various lipids) appear to be altered in AD independent of these risk factors[39,41,42]. In patients with MCI, alterations in lipid metabolism, lysine metabolism, and the tricarboxylic acid cycle have been observed[43,44]. In one of the largest blood-based metabolomics studies of AD, we identified

metabolic alterations in various stages across the trajectory of the disease. For instance, higher levels of SMs and PCs were observed in early stages of AD as defined by abnormal cerebrospinal fluid (CSF) Aβ$_{1–42}$ levels, whereas intermediate changes, measured by CSF total tau, were correlated with increased levels of SMs and long-chain acylcarnitines[45]. Changes in brain volume and cognition, usually noted in later stages, were correlated with a shift in energy substrate utilization from fatty acids to amino acids, especially BCAAs. Other metabolomics studies have reported metabolic alterations in AD that support these findings, including alterations in PCs in AD[44,46–48] and sphingolipid transport and fatty-acid metabolism in MCI/AD compared with cognitively normal (CN) participants[49]. Higher blood concentrations of sphingolipid species were associated with disease progression and pathological severity at autopsy[50]. Metabolomics analysis of brain and blood tissue revealed that bile acids, important regulators of lipid metabolism and products of human–gut microbiome co-metabolism, were altered in AD[51,52] and associated with brain glucose metabolism and atrophy as well as CSF Aβ$_{1–42}$ and p-tau[53]. In most of these studies, sex and *APOE* ε4 genotype were used as covariates. Thus, these studies may have missed sex-specific associations between AD and metabolite levels or associations with opposite effect directions for the two sexes. Similarly, sex-by-*APOE* genotype interactions would have been masked.

Here, we examined the role of sex in the relationship between metabolic alterations and AD to elucidate possible metabolic underpinnings for the observed sexual dimorphism in AD susceptibility and severity. Using metabolomics data from 1517 participants of the Alzheimer's Disease Neuroimaging Initiative (ADNI) cohorts, we investigated how sex modifies the associations of representative A-T-N biomarkers[54,55] (A: CSF Aβ$_{1–42}$ pathology; T: CSF p-tau; N: region of interest (ROI)-based glucose uptake measured by [$^{18}$F] fluorodeoxyglucose-positron emission tomography (FDG-PET)) with 139 blood metabolites using stratified analyses and systematic comparison of effects between men and women. In downstream analyses, we inspected sex differences in metabolic effects on AD biomarkers for dependencies on *APOE* genotype, both by interaction analysis and sub-stratification.

## Results

**Investigating metabolic effect modulation by AD risk factors.** We used CSF biomarkers, FDG-PET imaging, and serum metabolomics data on 139 metabolites to investigate metabolic effects in relation to sex and AD and their interaction. Of 1517 ADNI participants, 1082 had CSF Aβ$_{1–42}$ and p-tau levels and 1143 had FDG-PET data available (Table 1). We included all individuals with respective data regardless of diagnostic classification, as we were interested in these three representatives of the A-T-N AD biomarker schema[54,55] as our main readouts. In this data set, there was no significant difference in the number of *APOE* ε4+/− participants between females and males ($P_{Fisher's\ exact} > 0.3$). Of the three AD biomarkers, only p-tau levels were significantly different between sexes in covariate-adjusted regression models ($P_{REG} = 0.01$ after adjustment for three tests) with slightly higher levels in females.

Previous studies consistently showed widespread metabolic sex differences, metabolic imprint of genetic variance in the *APOE* locus and significant associations between blood metabolites and AD biomarkers independent of (i.e., adjusted for) sex. The current study added specific examination of the following central questions (Supplementary Fig. 1): (i) are peripheral metabolic sex differences changed owing to presence of (probable) AD?, (ii) are metabolite associations with A-T-N biomarkers modified by sex?,

**Table 1 Characteristics of the 1517 ADNI participants included in this study.**

| | Global data set | CN | SMC | EMCI | MCI | AD |
|---|---|---|---|---|---|---|
| $N_{subjects}$ | 1517 | 362 | 93 | 270 | 490 | 302 |
| Sex (m/f) | 828/689 | 177/185 | 39/54 | 149/121 | 298/192 | 165/137 |
| Age | 73.72 (+−7.25) | 74.61 (+−5.77) | 72.34 (+−5.70) | 71.26 (+−7.63) | 74.03 (+−7.63) | 74.79 (+−7.77) |
| BMI | 26.86(+− 4.82) | 26.99 (+−4.53) | 28.46 (+−6.23) | 27.96 (+−5.36) | 26.45 (+−4.27) | 25.88 (+−4.69) |
| Education | 15.88 (+−2.87) | 16.24 (+−2.79) | 16.78 (+−2.55) | 15.95 (+−2.67) | 15.84 (+−2.91) | 15.16 (+−3.01) |
| APOE ε4−/+ | 809/708* | 261/101 | 64/29 | 155/115 | 224/266 | 105/197 |
| CSF available | 1082* | 236 | 84 | 245 | 308 | 209 |
| Path. Aβ$_{1-42}$−/+ | 407/675 | 134/102 | 57/27 | 122/123 | 75/233 | 19/190 |
| CSF Aβ$_{1-42}$ | 1052.73 (+ −601.70) | 1324.60 (+ −652.13) | 1395.01 (+ −618.19) | 1172.73 (+ −569.12) | 896.35 (+ −501.80) | 697.95 (+ −431.49) |
| CSF p-Tau | 27.79 (+−14.56) | 22.01 (+−9.19) | 21.66 (+−9.14) | 24.34 (+−14.03) | 30.81 (+−14.94) | 36.38 (+−16.07) |
| FDG-PET available | 1143* | 247 | 93 | 268 | 318 | 217 |
| FDG-PET | 6.17 (+−0.77) | 6.53 (+−0.58) | 6.60 (+−0.58) | 6.44 (+−0.60) | 6.08 (+−0.68) | 5.36 (+−0.73) |
| *Numbers for combined stratification: | | | | | | |
| | APOE ε4− females | APOE ε4− males | APOE ε4+ females | APOE ε4+ males | | |
| Total | 374 | 435 | 315 | 393 | | |
| CSF available | 267 | 315 | 222 | 278 | | |
| FDG-PET available | 278 | 337 | 230 | 298 | | |

CN cognitively normal, SMC subjective memory complaints, EMCI early mild cognitive impairment, MCI mild cognitive impairment, AD probable Alzheimer's disease, BMI body mass index, APOE ε4−/+ non-carriers and carriers of the APOE ε4 allele, Path. Aβ$_{1-42}$−/+ participants who have normal and pathological CSF Aβ$_{1-42}$ levels, respectively.

and (iii) is there evidence for APOE ε4 status influencing metabolite associations with A-T-N biomarkers that show differences between sexes?

**No significant change of metabolic sex differences in AD**. We tested whether sex-associated differences in blood metabolite levels differ between patients with probable AD, participants with late MCI, and CN participants in the ADNI cohorts. In the complete cohort ($n = 1517$), we found 108 of 139 metabolites to be significantly associated with sex after multiple testing correction while adjusting for age, body mass index (BMI), ADNI study phase, and diagnostic group. Seventy of these associations replicate previous findings in a healthy population using a prior version of the same metabolomics platform[25] that provides measurements on 92 out of the 108 metabolites identified in ADNI. All SMs and the majority of PCs were more abundant in women. The majority of biogenic amines, amino acids, and acylcarnitines were more abundant in men.

Stratifying participants by diagnostic group revealed that 53 of the 108 metabolites showing significant sex differences were also significant in each of the three groups (AD, MCI, CN) alone, whereas 14 showed no significant difference in any group, probably owing to lower statistical power after stratification (Supplementary Data 1 and Supplementary Fig. 2). Significant sex differences limited to one diagnostic group were found for eight metabolites (PC aa C34:1, PC ae C34:3, PC ae C36:3, PC ae C36:4, PC ae C38:5, PC ae C40:5, histidine, C6/C4:1-DC) in patients with probable AD, for seven metabolites (C0, C3, C9, C18:2, SDMA, spermidine, t4-OH-Pro) in the MCI group, and for six metabolites (PC aa C42:0, PC ae C32:1, PC ae C42:3, SM(OH) C24:1, sarcosine, aspartate) in the CN group. All significant sex differences found were also significant in the full cohort. Comparisons of beta estimates for sex between AD and CN groups showed no significant effect heterogeneity, indicating reduced power as source for these observed differences (Supplementary Data 1). The only exception was PC aa C34:1, which showed significant heterogeneity ($P_{HET} = 0.029$) between AD patients compared with CN participants. Notably, in the larger healthy reference cohort, sex did not significantly affect the blood level of this metabolite when adjusting for the same study covariates (i.e., age and BMI)[25]. In summary, we did not find

evidence for sex differences of blood metabolite levels being significantly affected by presence of MCI or AD.

**Sex modifies associations of metabolites with AD biomarkers**. To investigate whether sex modifies the association between AD endophenotypes and metabolite concentrations, we tested for associations of the three representative A-T-N biomarkers, CSF Aβ$_{1-42}$ pathology, CSF p-tau levels, and brain glucose uptake measured via FDG-PET imaging, with concentrations of 139 blood metabolites. We did this in the full data set and separately in each sex using multivariable linear and logistic regression, followed by analysis of heterogeneity of effects between sexes. Table 2 lists the results for all metabolite–phenotype combinations and analyses of sex-by-metabolite interaction effects on A-T-N biomarkers that fulfilled at least one of the following criteria: (i) associations significant (at a Bonferroni threshold of $P_{REG} < 9.09 \times 10^{-4}$) in the full cohort; (ii) associations Bonferroni significant in one sex; (iii) associations that showed suggestive significance ($P_{REG} < 0.05$) in one sex coupled with significance for effect heterogeneity between female and male effect estimates. Results for all metabolites, phenotypes, and statistical models are provided in Supplementary Data 2. Systematic comparison of estimated effects in men and women for all metabolites is shown in Fig. 1. Based on this comparison, we classified metabolite–A-T-N biomarker associations into homogeneous effects if metabolites showed very similar effects in their association to the biomarker for both sexes (i.e., estimated heterogeneity $p$ value $P_{HET} > 0.05$), and heterogeneous effects if metabolites showed different effects in both sexes leading to significant heterogeneity (i.e., estimated $P_{HET} < 0.05$) and/or sex–metabolite interaction (for more details on the concept of effect heterogeneity, see Supplementary Fig. 3). Effects that were Bonferroni significant in only one sex with either significant effect heterogeneity between males and females or significant sex–metabolite interaction were considered sex-specific.

**Homogeneous effects**. Metabolites with homogeneous effects lie on or close to the diagonal going through the first and third quadrant when plotting the effect estimates in women against those in men (Fig. 1). We identified eight significant homogenous metabolite–phenotype associations with A-T-N biomarkers: CSF

**Table 2 Association results and heterogeneity estimates for metabolites in relation to A-T-N biomarkers stratified by sex.**

| Biomarker/ metabolite | Pooled effect | Pooled p value | Effect type | Males effect | Males p value | Females effect | Females p value | Sex diff. statistic | Sex diff. p value | Sex diff. $I^2$ | Interaction p value |
|---|---|---|---|---|---|---|---|---|---|---|---|
| *Pathological CSF Aβ$_{1-42}$* | | | | | | | | | | | |
| PC ae C44:6 | 0.283 | 2.58E-04 | Homogeneous | 0.282 | 5.96E-03 | 0.299 | 1.33E-02 | −0.107 | 9.15E-01 | 0.000 | 8.09E-01 |
| PC ae C44:4 | 0.265 | 4.57E-04 | Homogeneous | 0.274 | 6.29E-03 | 0.255 | 3.07E-02 | 0.119 | 9.05E-01 | 0.000 | 7.83E-01 |
| PC ae C44:5 | 0.260 | 5.23E-04 | Homogeneous | 0.294 | 3.26E-03 | 0.214 | 6.38E-02 | 0.522 | 6.01E-01 | 0.000 | 4.34E-01 |
| Threonine | 0.207 | 6.72E-03 | Male-specific | 0.372 | 8.83E-04 | 0.070 | 5.17E-01 | 1.943 | 5.20E-02 | 48.545 | 4.03E-02 |
| Valine | −0.134 | 1.05E-01 | Heterogeneous | 0.032 | 7.80E-01 | −0.299 | 1.50E-02 | 1.973 | 4.85E-02 | 49.322 | 7.65E-02 |
| *CSF p-tau* | | | | | | | | | | | |
| C10 | 0.084 | 4.58E-03 | Female-specific | 0.014 | 7.34E-01 | 0.144 | 6.07E-04 | −2.203 | 2.76E-02 | 54.613 | 2.55E-02 |
| C5−DC (C6−OH) | 0.103 | 2.35E-02 | Heterogeneous | 0.012 | 8.52E-01 | 0.205 | 2.27E-03 | −2.116 | 3.44E-02 | 52.740 | 3.38E-01 |
| C8 | 0.064 | 3.42E-02 | Heterogeneous | 0.003 | 9.39E-01 | 0.127 | 5.11E-03 | −2.028 | 4.26E-02 | 50.692 | 5.63E-02 |
| PC ae C36:2 | 0.056 | 8.65E-02 | Heterogeneous | 0.129 | 4.80E-03 | −0.023 | 6.18E-01 | 2.355 | 1.85E-02 | 57.535 | 2.16E-02 |
| Histidine | −0.034 | 2.72E-01 | Heterogeneous | 0.033 | 4.39E-01 | −0.105 | 1.97E-02 | 2.237 | 2.53E-02 | 55.290 | 2.42E-02 |
| Asparagine | 0.034 | 2.84E-01 | Heterogeneous | 0.107 | 1.66E-02 | −0.052 | 2.32E-01 | 2.550 | 1.08E-02 | 60.788 | 2.16E-02 |
| SM (OH) C16:1 | 0.032 | 3.10E-01 | Heterogeneous | 0.091 | 3.36E-02 | −0.039 | 3.99E-01 | 2.066 | 3.89E-02 | 51.592 | 3.75E-02 |
| Glycine | 0.030 | 3.50E-01 | Heterogeneous | 0.104 | 3.94E-02 | −0.026 | 5.23E-01 | 2.014 | 4.40E-02 | 50.346 | 6.88E-02 |
| PC ae C36:1 | 0.028 | 3.68E-01 | Heterogeneous | 0.088 | 4.17E-02 | −0.041 | 3.51E-01 | 2.094 | 3.62E-02 | 52.251 | 3.76E-02 |
| C2 | 0.015 | 5.85E-01 | Heterogeneous | −0.054 | 1.67E-01 | 0.089 | 3.02E-02 | −2.527 | 1.15E-02 | 60.430 | 1.39E-02 |
| *FDG-PET* | | | | | | | | | | | |
| PC aa C32:1 | −0.127 | 2.32E-05 | Homogeneous | −0.140 | 6.31E-04 | −0.110 | 1.50E-02 | −0.499 | 6.18E-01 | 0.000 | 5.53E-01 |
| PC ae C44:4 | −0.111 | 2.27E-04 | Homogeneous | −0.097 | 1.80E-02 | −0.141 | 1.84E-03 | 0.716 | 4.74E-01 | 0.000 | 2.21E-01 |
| PC ae C44:5 | −0.105 | 4.07E-04 | Homogeneous | −0.112 | 5.80E-03 | −0.111 | 1.30E-02 | −0.021 | 9.83E-01 | 0.000 | 6.02E-01 |
| PC aa C32:0 | −0.107 | 6.85E-04 | Homogeneous | −0.125 | 5.67E-03 | −0.091 | 4.25E-02 | −0.547 | 5.84E-01 | 0.000 | 7.44E-01 |
| PC ae C42:4 | −0.103 | 8.56E-04 | Homogeneous | −0.103 | 1.58E-02 | −0.112 | 1.33E-02 | 0.156 | 8.76E-01 | 0.000 | 4.48E-01 |
| C16:1 | −0.103 | 9.09E-04 | Male-specific | −0.165 | 9.64E-05 | −0.029 | 5.38E-01 | −2.179 | 2.93E-02 | 54.107 | 9.94E-02 |
| PC ae C40:2 | −0.053 | 7.82E-02 | Heterogeneous | −0.119 | 4.34E-02 | 0.016 | 7.15E-01 | −2.238 | 2.52E-02 | 55.312 | 5.78E-02 |
| Proline | −0.023 | 4.51E-01 | Heterogeneous | 0.059 | 1.77E-01 | −0.118 | 8.18E-03 | 2.841 | 4.50E-03 | 64.801 | 7.74E-03 |

Metabolite associations with A-T-N biomarkers that are either Bonferroni-significant in the full sample, in one sex, or show nominal significance both in one sex and for effect heterogeneity or the sex×metabolite interaction term. Given are regression results for the full sample and both sexes, as well as heterogeneity estimates and the p value for sex×metabolite interactions. Full association results for all strata are provided in Supplementary Data 2. *Sex diff.* results from the heterogeneity analysis.

Aβ$_{1-42}$ pathology was significantly associated with levels of three related ether-containing PCs (PC ae C44:4, PC ae C44:5, PC ae C44:6). Two of those (PC ae C44:4, PC ae C44:5) were also significantly associated with brain glucose uptake (FDG-PET) in addition to three other PCs (PC aa C32:1, PC aa C32:0, PC ae C42:4). For p-tau, we found no homogeneous, overall significant associations. Notably, none of the homogeneous associations showed any indication of effect heterogeneity between sexes, and only one reached significance in the sex-stratified analyses: higher blood levels of the diacyl-PC PC aa C32:1 were associated with lower glucose uptake in brain in the male stratum alone despite lower power.

**Heterogeneous effects**. Metabolites with heterogeneous effects fall mainly into the second or fourth quadrant (with the exception of sex-specific effects that are close to one of the axes) when contrasting the effect estimates for men and women in the plots for the three A-T-N phenotypes (Fig. 1). We identified 15 associations in this category (including three sex-specific effects). For CSF Aβ$_{1-42}$, we identified two heterogeneous effects, with threonine showing a sex-specific effect (see paragraph below) with greater effect size in males, and valine with a larger effect in females. Although valine was not significantly associated ($P_{REG} = 0.78$) with CSF Aβ$_{1-42}$ pathology in males, in females it showed a nominally significant negative association with an estimated heterogeneity of $I^2 = 49.3\%$. CSF p-tau had the largest number of heterogeneous associations: acylcarnitines C5-DC (C6-OH), C8, C10 (sex-specific), and C2, and the amino acid histidine showed stronger associations in females, whereas the related ether-containing PCs, PC ae C36:1 and PC ae C36:2, the amino acids asparagine and glycine, and one hydroxy-SM (SM (OH) C16:1) yielded stronger associations in males (all $I^2 > 50\%$). Associations with FDG-PET revealed three heterogeneous effects, with ether-containing PC ae C40:2 and the acylcarnitine C16:1 (sex-specific) showing a larger effect in males ($I^2 = 55.3$ and 54.1%, respectively), and proline having a larger effect in females ($I^2 = 64.8\%$). For seven of the 15 reported heterogeneous effects, the interaction term (sex × metabolite) in the full cohort was also significantly (at $P_{REG} < 0.05$) associated with the respective biomarker.

**Sex-specific effects**. Metabolites with sex-specific effects fall into the area close to the *x*- (male-specific) or *y*- (female-specific) axes of the three effect plots for the different A-T-N phenotypes (Fig. 1). We found three instances of this effect type. Male-specific effects were seen for threonine with pathological CSF Aβ$_{1-42}$ (positive association) and C16:1 with FDG-PET (negative association). One female-specific effect was seen: higher levels of the medium-chain acylcarnitine C10 were associated with higher CSF p-tau. This association was the strongest seen for p-tau in the full cohort analysis, yet seems to be driven by female effects only.

**Intertwined modulation of metabolite effects by sex and *APOE***. Previous reports suggested that the *APOE* ε4 genotype may exert AD risk predisposition in a sex-dependent way[8–13]. To investigate potential relationships between sex and *APOE* ε4 status on the metabolomic level, we selected the 21 metabolites identified in the previous analyses (Table 2) and performed association analyses with the three selected A-T-N biomarkers, now stratified by *APOE* ε4 status and adjusted for sex. Metabolite effects in *APOE* ε4 carriers vs. non-carriers showed effects from all three categories defined above (Table 3): Homogeneous effects were noted for the overall significant associations of PC aa C32:1, PC ae C44:4, PC ae C44:5, PC aa C32:0, and PC ae C42:4 with FDG-PET. Heterogeneous effects again formed the largest group ($n = 11$). Proline and glycine showed significantly different effects on CSF Aβ$_{1-42}$ pathology for ε4 carriers vs. non-carriers. For FDG-PET, significant heterogeneity between carriers and non-carriers was observed for C8, valine, glycine, and proline. Five metabolites with heterogeneous effects showed *APOE* ε4 status-specific effects: PC ae C44:6, PC ae C44:4, PC ae C44:5, and PC ae C42:4 showed Bonferroni-significant associations with pathological CSF Aβ$_{1-42}$ in *APOE* ε4 carriers; in the case of PC ae C44:6, PC ae C44:5, and PC ae C44:4, the group-specific effects were strong enough to drive the signal to overall significance in the full sample. Acylcarnitine C10 showed a Bonferroni-significant association with FDG-PET in *APOE* ε4 non-carriers.

**Some metabolic effects are specific to female ε4 carriers**. When we stratified separately by sex and *APOE* ε4 status, several

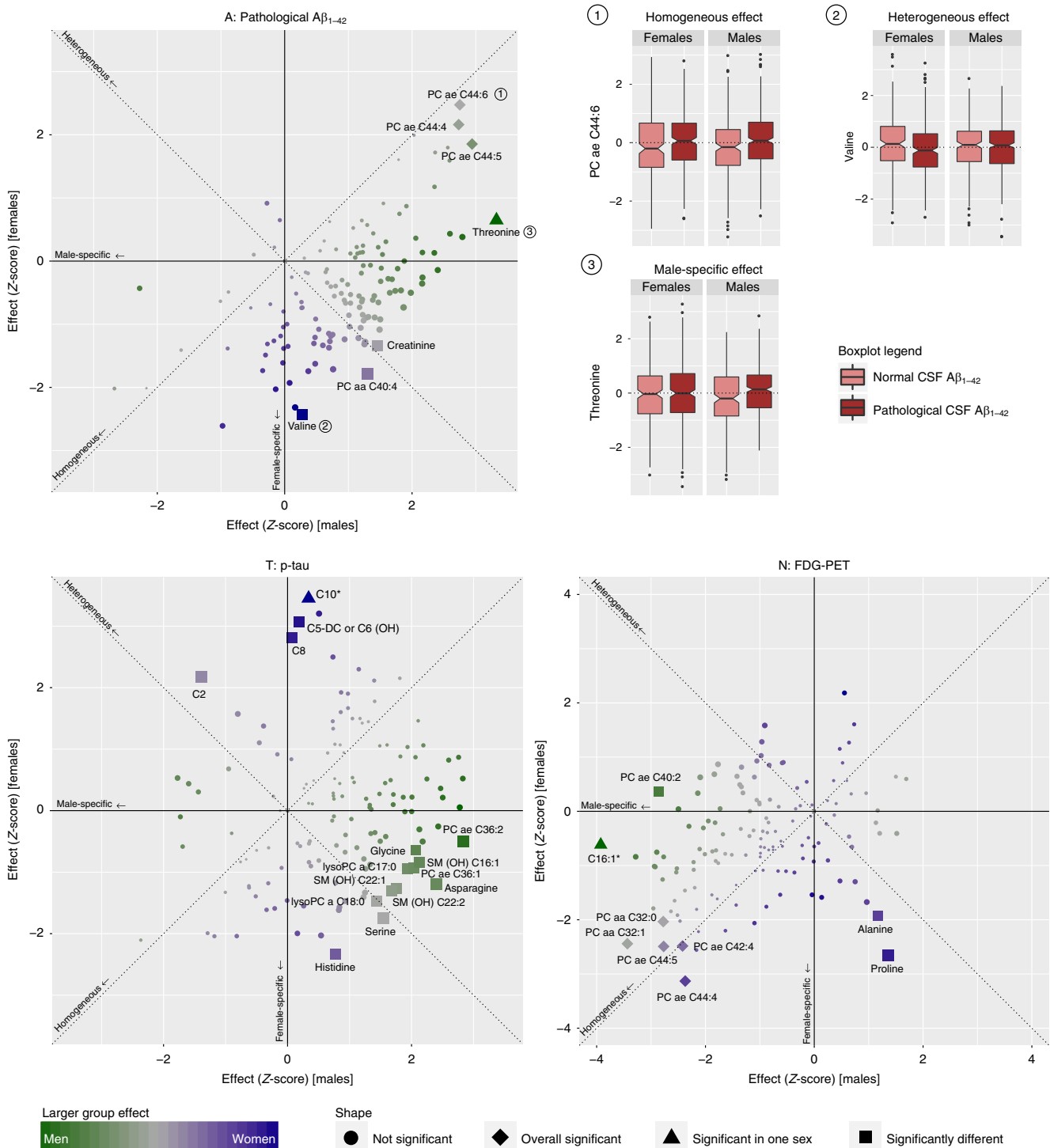

**Fig. 1 Sex-based effect heterogeneity of metabolites in relation to A-T-N biomarkers.** Scatter plots showing $Z$ scores of effect estimates of metabolite associations with A-T-N biomarkers for males ($x$ axis) versus females ($y$ axis). Homogeneous effects (i.e., those with same effect direction and comparable effect size) are located close to the diagonal, heterogeneous effects are located close to the anti-diagonal, and sex-specific effects are located close to the $x$ axis for male-specific and $y$ axis for female-specific effects. Homogeneous overall significant results are drawn as diamonds, effects with significant heterogeneity are drawn as rectangles, and effects significant in only one sex are drawn as triangles. Metabolites additionally marked by an asterisk are significant in one sex only and simultaneously show significant heterogeneity. Sex-specificity is further illustrated by a color scale (blue: females; green: males). On the upper right panel, example boxplots of metabolite residuals (obtained by regressing out included covariates) for each effect type are shown separately for females and males with (in dark red) and without (in light red) CSF A$\beta_{1-42}$ pathology, respectively. Source data are provided as Source Data File.

**Table 3 Association results and heterogeneity estimates for metabolites in relation to A-T-N biomarkers stratified by *APOE* ε4 status.**

| Biomarker/metabolite | Pooled effect | Pooled p value | Effect type | ε4+effect | ε4+p value | ε4− effect | ε4− p value | ε4 diff. statistic | ε4 diff. p value | ε4 diff. I² | Interaction p value |
|---|---|---|---|---|---|---|---|---|---|---|---|
| *Pathological CSF Aβ₁₋₄₂* | | | | | | | | | | | |
| PC ae C44:6 | 0.283 | 2.58E-04 | Specific to ε4+ | 0.630 | 2.50E-05 | 0.158 | 7.96E-02 | −2.705 | 6.83E-03 | 63.030 | 2.80E-03 |
| PC ae C44:4 | 0.265 | 4.57E-04 | Specific to ε4+ | 0.565 | 1.30E-04 | 0.139 | 1.13E-01 | −2.478 | 1.32E-02 | 59.645 | 5.80E-03 |
| PC ae C44:5 | 0.260 | 5.23E-04 | Specific to ε4+ | 0.609 | 2.64E-05 | 0.129 | 1.37E-01 | −2.837 | 4.56E-03 | 64.749 | 3.07E-03 |
| PC ae C42:4 | 0.242 | 1.98E-03 | Specific to ε4+ | 0.564 | 1.32E-04 | 0.114 | 2.15E-01 | −2.589 | 9.61E-03 | 61.382 | 5.64E-03 |
| Proline | −0.075 | 3.52E-01 | Heterogeneous | 0.176 | 2.15E-01 | −0.202 | 4.40E-02 | −2.173 | 2.98E-02 | 53.982 | 1.58E-01 |
| Glycine | 0.060 | 4.60E-01 | Heterogeneous | 0.363 | 1.83E-02 | −0.102 | 3.05E-01 | −2.538 | 1.11E-02 | 60.604 | 7.89E-04 |
| *FDG-PET* | | | | | | | | | | | |
| PC aa C32:1 | −0.127 | 2.32E-05 | Homogeneous | −0.087 | 5.35E-02 | −0.162 | 1.34E-04 | −1.210 | 2.26E-01 | 17.332 | 3.58E-01 |
| PC ae C44:4 | −0.111 | 2.27E-04 | Homogeneous | −0.115 | 1.39E-02 | −0.114 | 6.34E-03 | 0.023 | 9.82E-01 | 0.000 | 8.63E-01 |
| PC ae C44:5 | −0.105 | 4.07E-04 | Homogeneous | −0.122 | 8.34E-03 | −0.102 | 1.30E-02 | 0.326 | 7.44E-01 | 0.000 | 6.39E-01 |
| PC aa C32:0 | −0.107 | 6.85E-04 | Homogeneous | −0.135 | 4.39E-03 | −0.082 | 6.58E-02 | 0.818 | 4.14E-01 | 0.000 | 3.69E-01 |
| PC ae C42:4 | −0.103 | 8.56E-04 | Homogeneous | −0.131 | 5.79E-03 | −0.086 | 4.51E-02 | 0.701 | 4.83E-01 | 0.000 | 3.98E-01 |
| C10 | −0.057 | 5.14E-02 | Specific to ε4− | 0.037 | 4.17E-01 | −0.135 | 7.17E-04 | −2.840 | 4.51E-03 | 64.793 | 4.96E-03 |
| C8 | −0.051 | 9.96E-02 | Heterogeneous | 0.038 | 4.04E-01 | −0.138 | 1.58E-03 | −2.794 | 5.20E-03 | 64.215 | 6.37E-03 |
| Valine | 0.036 | 2.49E-01 | Heterogeneous | −0.040 | 4.08E-01 | 0.106 | 1.68E-02 | 2.234 | 2.55E-02 | 55.233 | 9.50E-02 |
| Glycine | −0.032 | 3.00E-01 | Heterogeneous | −0.140 | 3.05E-03 | 0.059 | 1.80E-01 | 3.092 | 1.99E-03 | 67.653 | 3.29E-03 |
| Proline | −0.023 | 4.51E-01 | Heterogeneous | −0.100 | 3.39E-02 | 0.048 | 2.64E-01 | 2.324 | 2.01E-02 | 56.977 | 6.35E-02 |

Associations of metabolites identified in the sex-centric analysis with A-T-N biomarkers that are either Bonferroni-significant in the full sample, in *APOE* ε4+ or *APOE* ε4− participants, or show nominal significance both in one *APOE* ε4 status group and for effect heterogeneity or the *APOE* ε4 status×metabolite interaction. Given are regression results for the full sample and both *APOE* ε4 status groups, as well as heterogeneity estimates and the p value for *APOE* ε4 status×metabolite interactions. Full association results for all strata are provided in Supplementary Data 2. ε4 diff. results from the heterogeneity analysis.

**Table 4 Association results and heterogeneity estimates for metabolites in relation to A-T-N biomarkers in *APOE* ε4 carriers stratified by sex.**

| Biomarker/metabolite | Pooled effect | Pooled p value | Sex diff. p value | Sex diff. I² | ε4 diff. p value | ε4 diff. I² | ε4+ (m) effect | ε4+ (m) p value | ε4+ (f) effect | ε4+ (f) p value |
|---|---|---|---|---|---|---|---|---|---|---|
| *Pathological CSF Aβ₁₋₄₂* | | | | | | | | | | |
| PC ae C44:6 | 0.283 | 2.58E-04 | 9.15E-01 | 0.000 | 6.83E-03 | 63.03 | 0.463 | 1.68E-02 | 0.922 | 1.90E-04 |
| PC ae C44:5 | 0.26 | 5.23E-04 | 6.01E-01 | 0.000 | 4.56E-03 | 64.749 | 0.521 | 6.17E-03 | 0.761 | 8.29E-04 |
| PC ae C42:4 | 0.242 | 1.98E-03 | 7.58E-01 | 0.000 | 9.61E-03 | 61.382 | 0.42 | 3.15E-02 | 0.761 | 8.65E-04 |
| *CSF p-tau* | | | | | | | | | | |
| C10 | 0.084 | 4.58E-03 | 2.76E-02 | 54.613 | 6.16E-01 | 0 | -0.064 | 3.24E-01 | 0.264 | 1.21E-04 |
| *FDG-PET* | | | | | | | | | | |
| Proline | −0.023 | 4.51E-01 | 4.50E-03 | 64.801 | 2.01E-02 | 56.977 | 0.046 | 4.76E-01 | −0.272 | 8.22E-05 |

Significant metabolite effects in the combined stratification (sex by *APOE* ε4 status) on A-T-N biomarkers are driven by or limited to *APOE* ε4+ females. Given are regression results for the full sample, *APOE* ε4+ males, *APOE* ε4+ females, as well as heterogeneity estimates by sex and *APOE* ε4 status. The only metabolite showing effect heterogeneity for both stratification variables was proline in its association with FDG-PET values. Full association results for all strata are provided in Supplementary Data 2. Sex diff. results from the sex-stratified heterogeneity analysis, ε4 diff. results from the *APOE* ε4 status-stratified heterogeneity analysis, ε4+ (m)/ε4+ (f) results for *APOE* ε4+ males and females, respectively.

metabolites (C8, C10, valine, glycine, and proline) showed heterogeneous effects on AD biomarkers in both stratifications. To investigate the potential additional subgroup-specific effects, we combined the two stratifications and investigated the selected set of 21 metabolites for sex-by-*APOE* ε4 status effect modulations. Although the group of *APOE* ε4-carrying women was the smallest among the four strata (n = 315), all Bonferroni-significant associations were found in this subgroup (Table 4): higher levels of three ether-containing PCs (PC ae C42:4, PC ae C44:5, and PC ae C44:6) were associated with pathological CSF Aβ₁₋₄₂, higher acylcarnitine C10 was associated with higher CSF p-tau, and higher proline levels were associated with lower FDG-PET values (Fig. 2). The latter was not observed in any other performed analysis. In addition, we found significant ($P_{REG} < 0.05$) interaction effects between all metabolites except C10 and *APOE* ε4 status on their associated endophenotypes in females only.

**Estimates of effects and effect heterogeneity are stable.** To investigate the robustness of findings reported in this study, we performed 1000 bootstrap re-samplings for each A-T-N biomarker to generate simulated population-based effect distributions for all significant associations (Supplementary Data 3). Overall, the difference between effect estimates obtained in the three rounds of original analyses (pooled sample, onefold, and twofold stratification) and the respective average effect estimate across all bootstraps (i.e., the variability by means of estimated bias) was marginal. We also did not find any instance of an originally significant association (at $P_{REG} \leq 0.05$) where the bootstrap-t 95% confidence interval contained zero. This means that the simulated population effect as estimated by bootstrapping is unequal to zero, suggesting robustness of our reported findings. Further, 91.97% of simulated effect distributions were normally distributed ($P_{Shapiro-Wilk} > 0.05$). Bootstrapping replicated significance of associations at the respective p value thresholds and the expected (post hoc) power of ≥ 50% (for details, see Supplementary Data 3) with only three exceptions: estimated effect heterogeneity between sexes for the association of valine with pathological CSF Aβ₁₋₄₂ was, although on average (i.e., averaged across all 1000 samples) significant, only significant in 49.9% of bootstraps; the significant associations of PC ae C44:5 and PC ae C42:4 with pathological CSF Aβ₁₋₄₂ in *APOE* ε4-carrying females on average narrowly missed the Bonferroni-corrected significance threshold ($P_{REG} = 9.45 \times 10^{-4}$ and $P_{REG} = 9.77 \times 10^{-4}$, respectively), although both metabolites showed Bonferroni-significant p values in > 50% of bootstraps.

**Replication of results in independent cohorts.** To the best of our knowledge, ADNI is currently the only study of AD with data on

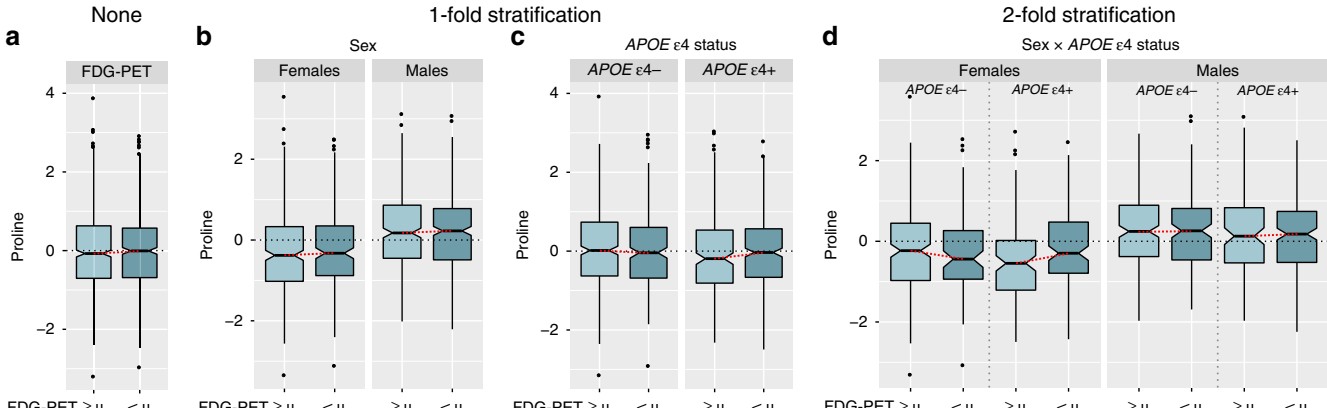

**Fig. 2 Group-specific association of proline levels with brain glucose uptake in *APOE* ε4-carrying females.** Boxplots showing residuals of proline levels (derived by regressing out covariate effects) for **a** the full sample; **b** onefold stratification by sex; **c** onefold stratification by *APOE* ε4 status; and **d** twofold stratification by both sex and *APOE* ε4 status; separately for high (light blue) and low (darker blue; derived by mean-split) FDG-PET values. The only subgroup showing a significant difference in proline levels are *APOE* ε4+ females with substantially higher levels in participants with lower brain glucose uptake. This Bonferroni-significant subgroup-specific effect ($P = 8.2 \times 10^{-5}$ in covariate-adjusted linear regression) would have been missed in sex/*APOE* ε4 genotype-adjusted analyses applying no or onefold stratification. Source data are provided as Source Data File.

both AD biomarkers and metabolite levels with sufficient sample sizes to conduct the reported analyses. Estimates of required sample sizes for replication of our findings are provided in Supplementary Table 1. We nevertheless sought independent replication of our results in two other studies with subsets of the examined variables available: (i) the Rush Religious Order Study and the Rush Memory and Aging Project (ROS/MAP)[56], for which we had access to 126 and 137 data points with data on p180 metabolites and data on overall amyloid load and severity of tau pathology in the brain (based on post-mortem neuropathology assessment), respectively (Supplementary Note 1). (ii) the Australian Imaging, Biomarker & Lifestyle Flagship Study of Ageing (AIBL) with data on CSF p-tau and comparable measurements of three lipid species (PC ae C36:1, PC ae C36:2, and SM (OH) C16:1) in 94 participants (Supplementary Note 2). Both studies had less than one quarter of the mean required sample size ($n = 677$).

We were able to replicate all homogeneous associations reported for pathological CSF Aβ$_{1-42}$ (PC ae C44:6, PC ae C44:5, and PC ae C44:4) in ROS/MAP at $p$ values significant after Bonferroni correction ($P_{REG} < 2.94 \times 10^{-3}$) and with the same effect directions as in ADNI, despite the different measure for Aβ pathology. For eight of the 14 sex- and *APOE* ε4 status-stratified associations for Aβ and tau pathology, we observed non-zero effect heterogeneity estimates ($I^2$ of 1.4–45.7%) (Supplementary Data 4), albeit non-significant. The three metabolite measures in AIBL all showed non-zero effect heterogeneity estimates ($I^2$ of 39.7–54.3%) in the sex-stratified analyses with CSF p-tau (Supplementary Data 5), with effect heterogeneity being significant for SM (OH) C16:1 ($P_{HET} = 0.016$). Combined, AIBL and ROS/MAP yielded non-zero heterogeneity estimates for two out of four reported group comparisons for Aβ pathology and eight out of ten reported group comparisons for CSF p-tau and brain tau pathology.

## Discussion

We investigated the influence of sex and *APOE* ε4 status on metabolic alterations related to representative A-T-N biomarkers (CSF Aβ$_{1-42}$ pathology (A), CSF p-tau (T), FDG-PET (N)). Using stratified analyses and systematic comparison of the effects estimated for the two sexes, we revealed substantial differences

between men and women in their associations of blood metabolite levels with these AD biomarkers, although known sexual dimorphisms of peripheral metabolite levels themselves were not significantly affected by MCI and probable AD status.

Differences between the sexes were largest for associations of metabolites and CSF p-tau levels. Notably, this biomarker was not significantly associated with any metabolite when including all participants and adjusting for both sex and copies of *APOE* ε4. However, association analysis stratified by sex (but still adjusted for copies of *APOE* ε4) revealed a significant, female-specific metabolite/CSF p-tau association despite the smaller sample size. For CSF Aβ$_{1-42}$ and FDG-PET, in addition to heterogeneous, sex-specific effects, we also found homogenous effects, in which metabolite concentrations showed the same trends of metabolite levels correlating with CSF Aβ$_{1-42}$ pathology and/or lower brain glucose uptake in both sexes.

For many of the metabolites with different effects for the sexes, we additionally observed significant effect heterogeneity between *APOE* ε4 carriers and non-carriers, suggesting intertwined modulation of metabolic effects by sex and *APOE* genotype. Indeed, twofold stratification revealed metabolite associations that were either driven by or specific to the group with presumably highest risk: *APOE* ε4-carrying females. Thus, our results demonstrate the importance of stratified analyses for getting insights into metabolic underpinnings of AD that are seemingly restricted to a specific patient group.

The metabolites showing effect heterogeneity across AD biomarkers in this study highlight sex-specific dysregulations of energy metabolism (acylcarnitines C2, C5-DC/C6-OH, C8, C10 and C16:1 for lipid-based energy metabolism[57]; amino acids valine, glycine, and proline as markers for glucogenic and ketogenic energy metabolism[58–60]), energy homeostasis (asparagine, glycine, proline, and histidine[59–63]), and (metabolic/nutrient) stress response (threonine, proline, histidine[60,62,64]). Although these pathways have been linked to AD before, our work presents the first evidence and molecular readouts for sex-related metabolic differences in AD.

For instance, our previous report discussed the implication of failing lipid energy metabolism in the context of AD biomarker profiles, starting at the stage of pathological changes in CSF tau levels[45]. The current study provides further insights into this topic, showing this finding to be predominant in females. More

specifically, we observed a significant female-specific association of higher levels of acylcarnitine C10 with increased levels of CSF p-tau, with two other metabolites of this pathway (C8 and C5-DC/C6-OH) narrowly falling short of meeting the Bonferroni threshold. This indicates a sex-specific buildup of medium-chain fatty acids in females, suggesting increased energy demands coupled with impaired energy production via mitochondrial beta-oxidation[57].

Interestingly, the significant heterogeneity of association shown by higher levels of glycine being linked to higher levels of CSF p-tau in men indicates that energy demands are equally upregulated in males as in females. However, men appear to compensate for this demand by upregulation of glucose energy metabolism as glycine is a positive marker of active glucose metabolism and insulin sensitivity[59]. Findings for acylcarnitines in females are further contrasted by the observed male-specific association of higher levels of the long-chain acylcarnitine C16:1 with decreased brain glucose uptake, which might indicate that in males there is a switch to provision of fatty acids as alternative fuel when glucose-based energy metabolism is less effective. As we did not observe the buildup of medium- and short-chain acylcarnitines as seen in females, we assume that, in males, energy production via mitochondrial beta-oxidation may be sustained, at least in early disease.

Evidence corroborating sex-specific processes in energy homeostasis linked to changes in CSF p-tau levels is provided by the significantly lower levels of histidine being linked to higher levels of CSF p-tau in women. Depletion of histidine is associated with insulin resistance, inflammatory processes, and oxidative stress, especially in women with metabolic dysregulation[61,62].

We further identified a heterogeneous association of valine with $A\beta_{1-42}$ pathology, with lower levels in females ($P_{REG} < 0.05$), but not in males. Valine, a BCAA and important energy carrying molecule, is associated with cognitive decline and brain atrophy in AD and risk for incident dementia[42,45]. The lower levels observed in AD are in contrast to other complex phenotypes such as type 2 diabetes, insulin resistance, or obesity[58,65], in which higher levels of BCAAs are found, and may indicate a switch to increased energy consumption via degradation of amino acids in AD. A recent study reported decreasing levels of valine being significantly associated with all-cause mortality[66]. Besides implications for energy metabolism, results from our study may thus characterize lower levels of valine also as a marker for increased female vulnerability to pathogenic processes in general and to β-amyloidosis in AD in particular.

To elaborate on potential interrelated risk predispositions via sex and *APOE* ε4 genotype from a metabolomics point of view, we further investigated whether *APOE* ε4 status may also modulate metabolic readouts of AD-linked A-T-N biomarker profiles identified in sex-centered analyses. Indeed, the majority (68.8%) of observed associations between metabolites and AD biomarkers showed significant heterogeneity between *APOE* ε4 status groups. Notably, the full set of metabolites yielding significant effect heterogeneity when comparing *APOE* ε4 carriers vs. non-carriers (C8, C10, glycine, proline, and valine) showed significant heterogeneity estimates in the sex-stratified analyses. We therefore applied twofold stratification by sex and *APOE* ε4 status to identify potential interactions between both variables (Supplementary Fig. 4). Several associations showed Bonferroni significance in the group with presumably the highest AD risk: *APOE* ε4+ females. One of the significant associations—higher proline levels with reduced brain glucose uptake—was not observed in the three other strata, the onefold stratifications, or the full sample, emphasizing the value of more fine-granular stratified analyses as proposed here.

The heterogeneity of metabolite effects as identified in this study might, in part, explain inconsistencies (e.g., ref. [67] vs. ref. [68]) in associations of metabolites and AD reported in other studies (e.g., if sex and *APOE* genotype are distributed differently and sample sizes are small). In contrast, the homogeneous effects reported in our study may represent more generic metabolic hallmarks in AD. For instance, PCs that presumably contain two long-chain fatty acids with, in total, four or five double bonds (PC ae C44:4, PC ae C44:5) were significantly associated with both CSF $A\beta_{1-42}$ and FDG-PET, showing homogeneous effects in males and females. These associations should be less sensitive to balancing of group sizes and replicate well across studies.

To test this assumption in an independent sample, we performed a targeted analysis using the three PCs homogeneously (PC ae C44:4, PC ae C44:5, PC ae C44:6) associated with CSF $A\beta_{1-42}$ pathology in 126 serum samples of 86 participants in the ROS/MAP cohorts (Supplementary Note 2). All three associations were Bonferroni significant with consistent effect directions, although we used a different measure of amyloid pathology as a proxy, namely total amyloid load in the brain, which is inversely correlated with CSF $A\beta_{1-42}$ levels[69]. This inverse relationship was mirrored by metabolite effect estimates. These results provide evidence for homogeneous associations to be relevant across cohorts.

There are several limitations to this study. First, the reported findings are observational and do not allow for direct causal conclusions. Second, the reported heterogeneity estimates still await replication in an independent cohort with sample sizes appropriate for stratification as well as metabolomics and endophenotypic data available. Available sample sizes in ROS/MAP and AIBL were too small to be sufficiently powered. Furthermore, ROS/MAP samples were poorly balanced for both sex (73% females) and *APOE* ε4 status (77% *APOE* ε4−), leading to even more limited power in the stratified analyses. Third, stratified analyses in combination with heterogeneity estimates may identify spurious associations, primarily due to the limited power resulting from group separation. However, bootstrapping analysis indicated overall robustness of our findings. In addition, we showed that for the majority of the non-homogeneous findings reported (60%), the interaction terms between metabolite levels and sex were also significant in the pooled analysis. When stratifying by *APOE* ε4 status, this was true for an even higher fraction of cases (72.7%). In conjunction with 71.4% of investigated effects showing non-zero heterogeneity in AIBL or ROS/MAP, this provides an additional line of support for the conclusions drawn in this work. Finally, we only investigated two AD risk factors using a limited metabolomics panel focused on lipid and amino acid metabolism. Further risk factors, such as type 2 diabetes, cardiovascular disease, and high blood pressure, are linked to metabolic aspects as well and may reveal even greater molecular heterogeneity, in particular, when expanding to additional metabolic pathways, e.g., by using non-targeted metabolomics.

In conclusion, effect heterogeneity between subgroups linked to energy metabolism reported in the present study has several important implications for AD research. First, this heterogeneity could explain inconsistencies of metabolomics findings between studies as observed for AD if participants showed different distributions of variables such as sex and *APOE* ε4 genotype. Second, pooled analysis with model adjustment for such variables, as typically applied for sex, can mask substantial effects that are relevant for only a subgroup of people. This is also true for combinations of stratifying variables as we demonstrated for the association of proline with brain glucose uptake in female *APOE* ε4 carriers. Consequently, drug trials may have more success by acknowledging between-group differences and targeting the

subgroup with the presumably largest benefit in their inclusion criteria. For energy metabolism, group-specific dietary interventions precisely targeting the respective dysfunctional pathways may pose a promising alternative to de novo drug development. Extending our approach by selection of additional variables to further improve stratification may eventually guide the way to precision medicine.

## Methods

**Study participants.** Data used in the preparation of this article were obtained from the ADNI database (http://adni.loni.usc.edu/). The ADNI was launched in 2003 as a public-private partnership. The primary goal of ADNI has been to test whether serial magnetic resonance imaging, PET, other biological markers, and clinical and neuropsychological assessment can be combined to measure the progression of MCI and early AD. For up-to-date information, see www.adni-info.org. Written informed consent was obtained at enrollment, which included permission for analysis and data sharing. Consent forms were approved by each participating site's institutional review board. Metabolomics data and results have been made accessible through the AMP-AD Knowledge Portal (https://ampadportal.org). The AMP-AD Knowledge Portal is the distribution site for data, analysis results, analytical methodology, and research tools generated by the AMP-AD Target Discovery and Preclinical Validation Consortium and multiple Consortia and research programs supported by the National Institute on Aging. Information on data availability and accessibility is available in the Data availability section.

We included 1517 baseline serum samples of fasting participants pooled from ADNI phases 1, GO, and 2. Demographics, diagnostic groups, and numbers and distributions of key risk factors are provided in Table 1. AD dementia diagnosis was established based on the NINDS-ADRDA criteria for probable AD for individuals with Mini-Mental State Exam (MMSE) scores between 20 and 26 (inclusive) and a Clinical Dementia Rating Scale (CDR) of 0.5 or 1.0. MCI participants did not meet these AD criteria and had MMSE scores between 24 and 30 (inclusive), a memory complaint, objective memory loss measured by education-adjusted scores on Wechsler Memory Scale Logical Memory II, a CDR of 0.5, absence of significant levels of impairment in other cognitive domains, and essentially preserved activities of daily living, meeting predetermined criteria for amnestic MCI[70]. Of the 1517 participants, 689 were female and 828 male, with 708 APOE ε4 carriers and 809 non-carriers. In the combined stratification by sex and APOE ε4 status (APOE ε4− = 0 copies of ε4, APOE ε4+ = 1 or 2 copies of ε4), the APOE ε4 non-carriers were separated into 374 females and 435 males, whereas of APOE ε4 carriers 315 were female and 393 male.

**Metabolomics data acquisition.** Metabolites were measured with the targeted AbsoluteIDQ-p180 metabolomics kit (BIOCRATES Life Science AG, Innsbruck, Austria), with an ultra-performance liquid chromatography tandem mass spectrometry (MS/MS) system (Acquity UPLC (Waters), TQ-S triple quadrupole MS/MS (Waters)), which provides measurements of up to 186 endogenous metabolites. Sample extraction, metabolite measurement, identification, quantification, and primary quality control (QC) followed standard procedures[45,71].

**Metabolomics data processing.** Metabolomics data processing followed published protocols[45,71] with a few adjustments. Raw metabolomics data for 182 metabolites was available for serum study samples of 1681 participants (four metabolites were removed owing to technical issues during measurement) measured on 23 plates. For each plate, 2–3 NIST Standard Reference samples were available. Furthermore, we also had blinded duplicated measurements for 19 samples (ADNI-1) and blinded triplicated measurements for 17 samples (ADNI-GO and -2) distributed across plates. We first excluded 23 metabolites with large numbers of missing values (>20%). Then, we removed plate batch effects applying cross-plate mean normalization using NIST metabolite concentrations (Supplementary Fig. 5). Duplicated and triplicated study samples were then utilized to calculate the coefficients of variation (exclusion criterion > 20%) and intra-class correlation (exclusion criterion < 65%) for each metabolite. We removed 20 metabolites that violated these thresholds. Next, we average-combined biological replicates and excluded non-fasting participants ($n = 108$), imputed missing metabolite data using half the value of the lower limit of detection per metabolite and plate, log2-transformed metabolite concentrations, centered and scaled distributions to a mean of zero and unit variance and winsorized single outlying values to three standard deviations. We then used the Mahalanobis distance for detection of multivariate subject outliers, applying the critical Chi-square value for $P < 0.01$ and removing 42 participants. Finally, metabolites were adjusted for significant medication effects using stepwise backwards selection (for details, see ref. [71]). The final QC-ed metabolomics data set was further restricted to individuals with data on all significant covariates (see section Phenotype data and covariate selection), resulting in the study data set of 139 metabolites and 1517 individuals. Derived quality control measures for all metabolites are provided in Supplementary Data 6, and their distributions after QC are provided in Supplementary Fig. 6.

**Phenotype data and covariate selection.** We limited association analyses of metabolites with AD to early detectable endophenotypes, specifically to the pathological threshold for CSF $A\beta_{1-42}$, levels of phosphorylated tau protein in the CSF (p-tau), and brain glucose metabolism measured by FDG-PET. Baseline data on these biomarkers for ADNI-1, -GO, and -2 participants were downloaded from the LONI online portal at https://ida.loni.usc.edu/. For CSF biomarker data, we used the data set generated by the validated and highly automated Roche Elecsys electrochemiluminescence immunoassays[72,73]. For FDG-PET, we used an ROI-based measure of average glucose uptake across the left and right angular, left and right temporal, and bilateral posterior cingulate regions derived from preprocessed scans (co-registered, averaged, standardized image and voxel size, uniform resolution) and intensity-normalized using a pons ROI to obtain standard uptake value ratio means[74,75]. The pathological CSF $A\beta_{1-42}$ cut-point (1073 pg/ml) as reported by the ADNI biomarker core for diagnosis-independent mixture modeling (see http://adni.loni.usc.edu/methods/, accessed Oct 2017) was used for categorization, as CSF $A\beta_{1-42}$ concentrations were not normally distributed. Processed FDG-PET values were scaled and centered to zero mean and unit variance prior to association analysis, p-tau levels were additionally log2-transformed. We extracted covariates including age, sex, BMI (calculated using baseline weight and body height), number of copies of the APOE ε4 genotype, and years of education. Covariates were separated into forced-in covariates (age, sex, ADNI study phase, and number of copies of APOE ε4) and covariates (BMI, education) selectable by backwards selection. ADNI study phase was included to adjust for remaining metabolic differences between batches (ADNI-1 and ADNI-GO/-2 were processed in separate runs) and differences in PET imaging technologies.

**Association analyses.** Association analyses of the three AD biomarkers with metabolite levels were conducted using standard linear (p-tau, FDG-PET) and logistic (pathological $A\beta_{1-42}$) regression. For pathological CSF $A\beta_{1-42}$, only BMI was additionally selected; for p-tau and FDG-PET, the full set of covariates was used. The stratification variables sex and copies of APOE ε4 were excluded as covariates in the sex-stratified and APOE ε4+/− status-stratified analyses, respectively. For identifying metabolic sex differences, we used linear regression with metabolite levels as the dependent variable and age, sex, BMI, ADNI study phase, and diagnostic group as explanatory variables and retrieved statistics for sex. To adjust for multiple testing, we accounted for the significant correlation structure across the 139 metabolites using the method of Li and Ji[76] and determined the number of independent metabolic features (i.e., tests) to be 55, leading to a threshold of Bonferroni significance of $9.09 \times 10^{-4}$. To assess significance of heterogeneity between strata, we used the methodology of[25,77] that is similar to the determination of study heterogeneity in inverse-weighted meta-analysis. We further provide a scaled (0–100%) index of percent heterogeneity similar to the $I^2$ statistic[78].

**Bootstrapping analysis.** Bootstrapping was performed using ordinary nonparametric bootstraps for each of the three A-T-N biomarkers separately. For this, we drew random indices with replacement a 1000 times from all participants in ADNI with the biomarker available. Association analysis was performed on each bootstrap using the same regression models as described above. We then calculated the bias of the effect estimates (i.e., the difference between effect estimates obtained in the original analyses and the respective average effect estimate across all bootstraps), as well as the bootstrap-$t$ (or studentized) 95% confidence interval that is taking into account the variance of the estimates in each single bootstrap. Averaged bootstrap statistics were obtained using the mean of the beta estimates and the mean of their standard errors across the set of 1000 bootstraps and using their ratio as statistic to retrieve associated two-tailed $p$ values from the standard normal distribution.

**Power analysis.** In each power analysis, we transformed covariate-adjusted effect sizes to sample size-weighted standardized effects (Cohen's $d$). For metabolic sex differences, we calculated the power for two-sample $t$ tests to identify significant sexual dimorphisms for metabolites with the standardized effect sizes observed in the pooled ADNI samples at Bonferroni significance in CN participants, participants with MCI, and patients with probable AD. To obtain estimates of sample sizes required to replicate metabolite associations and heterogeneity estimates, we used the same approach with power fixed to 50% (the post hoc/observed power to find results at $p$ values equal or below the respective applied threshold, i.e., nominal or Bonferroni significance). Thereby, we estimated sample sizes assuming perfectly balanced data sets (with respect to sex and APOE ε4 status). This is a very rough approximation as it further assumes that the effect sizes reported for ADNI are generalizable to any replication cohort. Therefore, reported required sample sizes may deviate in reality.

**Replication analysis in ROS/MAP and AIBL.** The ROS/MAP studies are both longitudinal cohort studies of aging and AD at Rush University[56] and are designed to be used in joint analyses to maximize sample size. Both studies were approved by an Institutional Review Board of Rush University Medical Center. All participants signed an informed consent and a repository consent to allow their biospecimens and data to be used for ancillary studies. We measured metabolite levels using the

AbsoluteIDQ-p180 metabolomics kit in 596 serum samples from 559 participants (37 additional samples from follow-up visits). Brain amyloid pathology data were available for 89 participants (126 serum samples) comprised of 40 CN, 28 MCI, and 21 AD participants; 100 participants (137 serum samples) had brain tau pathology data (46 CN, 28 MCI, and 26 AD). Key demographic characteristics of ROS/MAP cohorts and information on data acquisition and processing are provided in Supplementary Note 1. To obtain maximal power for replication, we included longitudinal metabolomics data where available and applied linear mixed models for association analysis. We used the same covariates as in ADNI, including study phase (ROS or MAP), sex, age at visit, BMI, copies of *APOE* ε4, and education (only for tau pathology). Race was added as additional covariate. Random effects (intercept) in the mixed models were included for both visit and participant identifiers.

AIBL is a longitudinal study of over 1100 people assessed over > 4.5 years to determine which biomarkers, cognitive characteristics, and health and lifestyle factors determine subsequent development of symptomatic AD. The AIBL study was approved by the institutional ethics committees of Austin Health, St. Vincent's Health, Hollywood Private Hospital and Edith Cowan University, and all volunteers gave written informed consent before participating in the study. We had access to measurements of CSF p-tau for 94 participants (82 CN, 7 MCI, and 5 AD) with lipidomic data available. In contrast to ADNI, lipidomic data in AIBL was assessed on the UHPLC-MS/MS platform of the Metabolomics Laboratory of the Baker Heart and Diabetes Institute, Melbourne, Australia, and not the AbsoluteIDQ-p180 metabolomics kit. As a consequence, matching measures of only three metabolites (PC ae C36:1, PC ae C36:2, and SM (OH) C16:1) could be derived in AIBL and were available for replication. Details on the matching process, as well as key demographic characteristics of AIBL participants and information on data acquisition and processing are provided in Supplementary Note 2. Association analysis was performed for log-transformed CSF p-tau levels and the three metabolite measures using linear regression while adjusting for sex, age, BMI, *APOE* ε4 status, and education. For both ROS/MAP and AIBL, sex and *APOE* ε4, respectively, were omitted as covariates in stratified analyses and heterogeneity estimates were calculated as in ADNI.

**Reporting summary**. Further information on research design is available in the Nature Research Reporting Summary linked to this article.

## Data availability

Metabolomics data sets from the AbsoluteIDQ-p180 metabolomics kit used in the current analyses for the ADNI-1, ADNI-GO/-2, and ROS/MAP cohorts are available via the Accelerating Medicines Partnership-Alzheimer's Disease (AMP-AD) Knowledge Portal (https://doi.org/10.7303/syn2580853) and can be accessed at https://doi.org/10.7303/syn5592519 (ADNI-1), https://doi.org/10.7303/syn9705278 (ADNI-GO/-2), and https://doi.org/10.7303/syn10235592 (ROS/MAP). The full complement of clinical and demographic data for the ADNI cohorts are hosted on the LONI data sharing platform and can be requested at http://adni.loni.usc.edu/data-samples/access-data/. The full complement of clinical and demographic data for the ROS/MAP cohorts are available via the Rush AD Center Resource Sharing Hub and can be requested at https://www.radc.rush.edu. AIBL data are available upon request at https://aibl.csiro.au. Source data are provided as Source Data File.

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

## Acknowledgements

Data used in preparation of this article were obtained from the ADNI database (adni.loni.usc.edu). As such, the investigators within the ADNI contributed to the design and implementation of ADNI and/or provided data but did not participate in analysis or writing of this report. A complete listing of ADNI investigators can be found at: http://adni.loni.usc.edu/wp-content/uploads/how_to_apply/ADNI_Acknowledgement_List.pdf. Metabolomics data used in preparation of this article were generated by the Alzheimer's Disease Metabolomics Consortium (ADMC) and obtained from the AMP-AD Knowledge Portal (https://doi.org/10.7303/syn2580853). As such, the investigators within the ADMC other than named authors provided data but did not participate in analysis or writing of this report. A complete listing of ADMC investigators can be found at: https://sites.duke.edu/adnimetab/team/. Data collection and sharing for this project was funded by the ADNI (National Institutes of Health Grant U01 AG024904) and DOD ADNI (Department of Defense award number W81XWH-12-2-0012). ADNI is funded by the National Institute on Aging, the National Institute of Biomedical Imaging and Bioengineering, and through generous contributions from the following: AbbVie, Alzheimer's Association; Alzheimer's Drug Discovery Foundation; Araclon Biotech; BioClinica, Inc.; Biogen; Bristol-Myers Squibb Company; CereSpir, Inc.; Cogstate; Eisai Inc.; Elan Pharmaceuticals, Inc.; Eli Lilly and Company; EuroImmun; F. Hoffmann-La Roche Ltd and its affiliated company Genentech, Inc.; Fujirebio; GE Healthcare; IXICO Ltd.; Janssen Alzheimer Immunotherapy Research & Development, LLC.; Johnson & Johnson Pharmaceutical Research & Development LLC.; Lumosity; Lundbeck; Merck & Co., Inc.; Meso Scale Diagnostics, LLC.; NeuroRx Research; Neurotrack Technologies; Novartis Pharmaceuticals Corporation; Pfizer Inc.; Piramal Imaging; Servier; Takeda Pharmaceutical Company; and Transition Therapeutics. The Canadian Institutes of Health Research is providing funds to support ADNI clinical sites in Canada. Private sector contributions are facilitated by the Foundation for the National Institutes of Health (www.fnih.org). The grantee organization is the Northern California Institute for Research and Education, and the study is coordinated by the Alzheimer's Therapeutic Research Institute at the University of Southern California. ADNI data are disseminated by the Laboratory for Neuro Imaging at the University of Southern California. The Religious Orders and the Rush Memory and Aging studies were supported by the National Institute on Aging grants P30AG10161, R01AG15819, R01AG17917, U01AG46152, and U01AG61356. National Institute on Aging (NIA) supported this work (R01 AG059093). NIA also supported the Alzheimer Disease Metabolomics Consortium which is a part of NIA's national initiatives AMP-AD and M²OVE-AD (R01 AG046171, RF1 AG051550, and 3U01 AG024904-09S4). In addition, M.A., R.K.D., and G.K. are supported by NIA grants RF1 AG058942 and R01 AG057452. M.A. and G.K. are also supported by funding from Qatar National Research Fund NPRP8-061-3-011. K.N. is supported by NIH grants NLM R01 LM012535 and NIA R03 AG054936. X.H. is supported by NIA grant RF1 AG061872. P.M.D. is supported by the NIH, Cure Alzheimer's Fund and the Karen L. Wrenn Trust.

## Author contributions

Conceptualization, M.A., R.K.D., and G.K.; methodology, M.A. and G.K.; data analysis, M.A., K.N., R.C., and G.K.; replication analyses, M.A., D.A.B., J.K., K.H., and P.J.M; metabolomics data generation and QC, M.A., K.N., J.B.T., T.M., K.H., P.J.M, J.K., R.M., S.M., M.A.M., J.W.T., L.S.J.W., J.D.T., X.H., R.B. and C.B.; sample preparation and handling, G.L., M.A.M, J.W.T., L.S.J.W., J.Q.T., and L.M.S.; data management, C.B., J.D.T., and G.L.; resources, M.W.W., L.M.S., J.Q.T, R.M.; visualization, M.A., G.K.; supervision, R.K.D., A.J.S., M.A., G.K., R.M., P.J.M., D.A.B., and P.M.D.; funding acquisition, R.K.D., D.A.B., R.M., M.A., G.K., and P.J.M; writing—original draft, M.A., G.K., A.K.P., J.K., P.M.D., A.J.S., and R.K.D; results interpretation, M.A., G.K., B.B., R.D.B., R.B., X.H., E.T., P.M.D., A.J.S., and R.K.D.; writing—review and editing, all authors.

## Competing interests

P.M.D. has received research grants (through Duke University) from Avid/Lilly, Neuronetrix, Avanir, Salix, Alzheimer's Drug Discovery Foundation, DOD and NIH. P.M.D. has received speaking or advisory fees from Anthrotronix, Neuroptix, Genomind, Clearview, Verily, RBC, Brain Canada, and CEOs Against Alzheimer's. P.M.D. owns shares in Muses Labs, Anthrotronix, Evidation Health, Turtle Shell Technologies, and Advera Health whose products are not discussed here. P.M.D. served on the board of Baycrest and serves on the board of Apollo Hospitals. P.M.D. is a co-inventor (through Duke) on patents relating to dementia biomarkers, metabolomics, and therapies, which are unlicensed. R.K.D. is inventor on key patents in the field of metabolomics, including applications for Alzheimer disease. M.A., J.B.T., G.K., M.A.M., J.W.T., R.B., X.H., L.S.J.W., A.J.S., K.N. are co-inventors on patent WO2018049268 in this field. J.B.T. further reports investigator-initiated research support from Eli Lilly unrelated to the work reported here. J.Q.T. may accrue revenue in the future on patents submitted by the University of Pennsylvania wherein he is a co-inventor and he received revenue from the sale of Avid to Eli Lilly as a co-inventor on imaging-related patents submitted by the University of Pennsylvania. L.M.S. is a consultant for Eli Lilly, Novartis, and Roche; he provides QC oversight for the Roche Elecsys immunoassay as part of responsibilities for the ADNI3 study. A.J.S. reports investigator-initiated research support from Eli Lilly unrelated to the work reported here. He has received consulting fees and travel expenses from Eli Lilly and Siemens Healthcare and is a consultant to Arkley BioTek. He also receives support from Springer publishing as an editor-in-chief of Brain Imaging and Behavior. M.W.W. reports stock/stock options from Elan, Synarc, travel expenses from Novartis, Tohoku University, Fundacio Ace, Travel eDreams, MCI Group, NSAS, Danone Trading, ANT Congress, NeuroVigil, CHRU-Hopital Roger Salengro, Siemens, AstraZeneca, Geneva University Hospitals, Lilly, University of California, San Diego–ADNI, Paris University, Institut Catala de Neurociencies Aplicades, University of New Mexico School of Medicine, Ipsen, Clinical Trials on Alzheimer's Disease, Pfizer, AD PD meeting. All other authors declare no competing interests.

## Additional information

Matthias Arnold [1,2,26], Kwangsik Nho[3,26], Alexandra Kueider-Paisley[1], Tyler Massaro[4], Kevin Huynh[5], Barbara Brauner[2], Siamak MahmoudianDehkordi[1], Gregory Louie [1], M. Arthur Moseley[6], J. Will Thompson[6], Lisa St John-Williams[6], Jessica D. Tenenbaum [7], Colette Blach[8], Rui Chang[9], Roberta D. Brinton[9,10,11], Rebecca Baillie [12], Xianlin Han [13], John Q. Trojanowski [14], Leslie M. Shaw[14], Ralph Martins[15,16], Michael W. Weiner[17], Eugenia Trushina[18,19], Jon B. Toledo[14,20], Peter J. Meikle [5], David A. Bennett[21], Jan Krumsiek[22], P. Murali Doraiswamy[1,23,24], Andrew J. Saykin [3], Rima Kaddurah-Daouk [1,23,24] & Gabi Kastenmüller [2,25]

[1]Department of Psychiatry and Behavioral Sciences, Duke University, Durham, NC, USA. [2]Institute of Bioinformatics and Systems Biology, Helmholtz Zentrum München, German Research Center for Environmental Health, Neuherberg, Germany. [3]Department of Radiology and Imaging Sciences and the Indiana Alzheimer Disease Center, Indiana University School of Medicine, Indianapolis, IN, USA. [4]Duke Clinical Research Institute, Duke University, Durham, NC, USA. [5]Metabolomics Laboratory, Baker Heart and Diabetes Institute, Melbourne, VIC, Australia. [6]Duke Proteomics and Metabolomics Shared Resource, Center for Genomic and Computational Biology, Duke University, Durham, NC, USA. [7]Department of Biostatistics and Bioinformatics, Duke University, Durham, NC, USA. [8]Duke Molecular Physiology Institute, Duke University, Durham, NC, USA. [9]Center for Innovation in Brain Science, University of Arizona, Tucson, AZ, USA. [10]Department of Pharmacology, College of Medicine, University of Arizona, Tucson, AZ, USA. [11]Department of Neurology, College of Medicine, University of Arizona, Tucson, AZ, USA. [12]Rosa & Co LLC, San Carlos, CA, USA. [13]University of Texas Health Science Center at San Antonio, San Antonio, TX, USA. [14]Department of Pathology & Laboratory Medicine, University of Pennsylvania, Philadelphia, PA, USA. [15]School of Medical and Health Sciences, Edith Cowan University, Joondalup, WA, Australia. [16]Department of Biomedical Sciences, Macquarie University, North Ryde, NSW, Australia. [17]Center for Imaging of Neurodegenerative Diseases, Department of Radiology, San Francisco VA Medical Center/University of California San Francisco, San Francisco, CA, USA. [18]Department of Neurology, Mayo Clinic, Rochester, MN, USA. [19]Department of Molecular Pharmacology and Experimental Therapeutics, Mayo Clinic, Rochester, MN, USA. [20]Department of Neurology, Houston Methodist Hospital, Houston, TX, USA. [21]Rush Alzheimer's Disease Center, Rush University Medical Center, Chicago, IL, USA. [22]Institute for Computational Biomedicine, Englander Institute for Precision Medicine, Department of Physiology and Biophysics, Weill Cornell Medicine, New York, NY, USA. [23]Duke Institute of Brain Sciences, Duke University, Durham, NC, USA. [24]Department of Medicine, Duke University, Durham, NC, USA. [25]German Center for Diabetes Research (DZD), Neuherberg, Germany. [26]These authors contributed equally: Matthias Arnold, Kwangsik Nho. ✉email: kaddu001@mc.duke.edu; g.kastenmueller@helmholtz-muenchen.de

