## [Peer Review File · Nature Communications]

Reviewers' comments:

Reviewer #1 (Remarks to the Author):

Late-onset Alzheimer's disease (AD) is the most common form of dementia in the aged with higher prevalence in the females. The APOE4 allele has been well documented to be the strongest genetic risk factor for late-onset AD. The present study was designed to dissect the sex- and APOE4-dependent metabolic heterogeneity in late-onset AD by focusing on group-specific metabolic alterations via stratified association analyses.

Overall, the research addressed a very important issue in the field, and the results clearly demonstrate the sex- and APOE4-dependent metabolic heterogeneity. The findings may explain inconsistencies of metabolomics findings in the field of AD research, and emphasize the need and importance of dissecting metabolic heterogeneity in AD pathogenesis.

Reviewer #2 (Remarks to the Author):

Arnold et al. present a study to evaluate the metabolomic effect of Alzheimer's Disease. The authors employ serum samples to ascertain metabolic alterations in Alzheimer's Disease, mid cognitive impairment and control participants. To do so, the authors analyzed the data generated for the ADNI cohort, and analyze the effects of sex and APOE e4 on the metabolome and AD, by stratifying the samples. I found several analytical issues that I think the authors should consider:

- i) The study by its design, cannot provide any conclusive evidence of negative association, as the authors are analyzing plasma and not brain tissue. For example, I consider that the authors are outreaching in the section 3.1 Metabolic sex-differences are unaffected by MCI and probable AD status is outreaching.
- ii) No replication is presented. The significant associations are not validated in any independent cohort, or additional tissue. Furthermore, no cross-validation technique as employed to validate the findings
- iii) It's not clear how many analytes are testing. In the manuscript the authors mention In section 2.2 "Metabolite data acquisition" 186 metabolites. Then in section 2.3 "Metabolomics data processing", they reported 182 metabolites. In addition, the description of the panel has a higher number of metabolites.
- iv) The authors are presenting data of analytes that were imputed for up to 40% of the subjects, which seems a too high percentage of missing values.
- v) Do metabolites follow a normal distribution?
- vi) Power is mentioned as a potential reason for lack of significance for expected associations, but no power analyses/simulation is presented.
- vii) My understating is that the authors are comparing effects on males and females in Table 2, but some of the effects that are being compared are not significantly associated, indicating that effect can even change of direction.
- viii) Some of the results are already presented in Varma et al (2018) Plos Med.

Reviewer #3 (Remarks to the Author):

The major claims of the paper are the use of stratified association analyses on the metabolic aspects of AD, differencing between males and females and between APOE ε4 carriers and non-carriers. The results are novel and they would be of interest to other researchers in the AD field.

Some minor comments that I would like to be explained in detail are:

1. The use of targeted metabolomics kit with UPLC/MS/MS, instead of an untargeted analysis.

2. The removed plate batch effects using cross-plate mean normalization using NIST metabolite concentrations.

3. The different types of study subjects, specifically MCI, are not indicated in the methods section.

Response to Reviewer Comments

We thank the reviewers for their valuable comments. We incorporated their suggestions to our best ability and feel that our manuscript has greatly benefitted from the review process. Below, we provide a point-by-point response to the reviewers' comments.

Please be aware that we had to make some additional changes to the manuscript in order to format it according to the *Nature Communications* guidelines for authors (e.g., we moved the methods section to the end of the manuscript).

Reviewer #1 (Remarks to the Author):

Late-onset Alzheimer's disease (AD) is the most common form of dementia in the aged with higher prevalence in the females. The APOE4 allele has been well documented to be the strongest genetic risk factor for late-onset AD. The present study was designed to dissect the sex- and APOE4-dependent metabolic heterogeneity in late-onset AD by focusing on group-specific metabolic alterations via stratified association analyses.

Overall, the research addressed a very important issue in the field, and the results clearly demonstrate the sex- and APOE4-dependent metabolic heterogeneity. The findings may explain inconsistencies of metabolomics findings in the field of AD research, and emphasize the need and importance of dissecting metabolic heterogeneity in AD pathogenesis.

We thank the reviewer for this very positive assessment of our work.

Reviewer #2 (Remarks to the Author):

Arnold et al. present a study to evaluate the metabolomic effect of Alzheimer's Disease. The authors employ serum samples to ascertain metabolic alterations in Alzheimer's Disease, mid cognitive impairment and control participants. To do so, the authors analyzed the data generated for the ADNI cohort, and analyze the effects of sex and APOE e4 on the metabolome and AD, by stratifying the samples. I found several analytical issues that I think the authors should consider:

- i) The study by its design, cannot provide any conclusive evidence of negative association, as the authors are analyzing plasma and not brain tissue. For example, I consider that the authors are outreaching in the section 3.1 Metabolic sex-differences are unaffected by MCI and probable AD status is outreaching.

We fully agree with the reviewer. We globally revised the manuscript to emphasize that our findings are limited to peripheral metabolite levels only. Further, we agree that missing significance does not allow for final conclusions on the topic of metabolic sex differences and adjusted the language accordingly:

- The title of the respective section in the results now reads: *"Peripheral metabolic sex differences are not significantly affected by MCI and probable AD status"*.

- The conclusion of this paragraph reads: *“In summary, we did not find evidence for sex differences of blood metabolite levels being significantly affected by presence of MCI or AD.”*
- ii) No replication is presented. The significant associations are not validated in any independent cohort, or additional tissue. Furthermore, no cross-validation technique as employed to validate the findings

We thank the reviewer for this correct comment. As stated in the original submission, we were unable to identify a cohort study with all data and, more importantly, the sample sizes available to replicate our findings beyond the initially reported replication of homogeneous associations of metabolites with A β pathology.

This situation unfortunately has not changed during the revision of the article. We have contacted and reviewed available data for many of the present AD studies, including the Baltimore Study of Ageing, the Age, Gene/Environment Susceptibility Reykjavik Study, the Rhineland Study, ROS/MAP, the AIBL study, and studies of several Alzheimer’s Disease Research Centers. Yet, none of them had all variables (or, in some cases, any of the key variables) or the sample sizes available that would have been needed. This is particularly true for FDG-PET imaging data, for which we found no suitable replication cohort that also provides metabolomics data.

Nonetheless, we completely agree with the reviewer regarding the importance of validation and replication of results. Therefore, we tried to address this issue in the revised manuscript to the largest extent possible:

- A) Robustness: We performed thorough bootstrapping analysis in the ADNI (cross-validation was not applicable in this context) to test the robustness of our results. Thereby, we could show that the vast majority of our findings is very stable across 1000 random re-samplings. We added methods and results on these analyses for all reported findings to the manuscript and the supplement and clearly stated, for which of the presented results we found evidence for potential uncertainty. The respective excerpt from the revised results section is pasted in below:

“To investigate the robustness of findings reported in this study, we performed 1000 bootstrap re-samplings for each A-T-N biomarker to generate simulated population-based effect distributions for all significant associations (Supplementary File 1). Overall, the difference between effect estimates obtained in the three rounds of original analyses (pooled sample, 1-fold, and 2-fold stratification) and the respective average effect estimate across all bootstraps (i.e., the variability by means of estimated bias) was marginal. We also did not find any instance of an originally significant association (at $P \leq 0.05$) where the bootstrap-t 95% confidence interval contained zero. This means that the simulated population effect as estimated by bootstrapping is unequal zero, suggesting robustness of our reported findings. Further, 91.97% of simulated effect distribution were normally distributed (PShapiro-Wilk > 0.05). Bootstrapping replicated significance of associations at the respective p-value thresholds and the expected (post-hoc) power of $\geq 50\%$ (for details, see Supplementary File 1) with only three exceptions: estimated effect heterogeneity between sexes for the association of valine with pathological CSF A β 1-42 was, although on average (i.e.

averaged across all 1000 samples) significant, only significant in 49.9% of bootstraps; the significant associations of PC ae C44:5 and PC ae C42:4 with pathological CSF A β 1-42 in APOE ϵ 4-carrying females on average narrowly missed the Bonferroni-corrected significance threshold ($P = 9.45 \times 10^{-4}$ and $P = 9.77 \times 10^{-4}$, respectively), although both metabolites showed Bonferroni-significant P-values in >50% of bootstraps.”

- B) Replication: We attempted replication of our findings with the respective subset of available phenotypes (or proxies thereof) and metabolites in two studies, namely ROS/MAP and AIBL although sample sizes were likely to be insufficient. To illustrate this power issue, we added post-hoc power analysis, which we provided in Supplementary Table 3. For ROS/MAP, we used brain pathology measures as proxies for the CSF-based traits available in ADNI and performed replication analysis. In AIBL, we had data on CSF p-tau levels available and metabolomics data from a metabolomics platform differing from the one that has been used in our study. Therefore, we had to derive comparable measures which unfortunately was only possible for three metabolic markers. Despite low power and imperfect data basis, we could replicate (at Bonferroni significance) all homogenous associations examined as well as one heterogeneity estimate. For the remaining 13 investigated heterogeneous associations, we found non-zero heterogeneity for 9 estimates, suggesting that these findings might be consistent across cohorts. The respective excerpt from the revised results section is pasted in below:

“To the best of our knowledge, ADNI is currently the only study of AD with data on both AD biomarkers and metabolite levels with sufficient sample sizes to conduct the reported analyses. Estimates of required sample sizes for replication of our findings are provided in Supplementary Table 3. We nevertheless sought independent replication of our results in two other studies with subsets of the examined variables available: (i) the Rush Religious Order Study and the Memory and Aging Project (ROS/MAP), for which we had access to 126 and 137 data points with data on p180 metabolites and data on overall amyloid load and severity of tau pathology in the brain (based on post-mortem neuropathology assessment), respectively (Supplementary Text 1). (ii) the Australian Imaging, Biomarker & Lifestyle Flagship Study of Ageing (AIBL) with data on CSF p-tau and comparable measurements of three lipid species (PC ae C36:1, PC ae C36:2, and SM (OH) C16:1) in 94 participants (Supplementary Text 2). Both studies had less than one quarter of the mean required sample size ($n = 677$).

We were able to replicate all homogeneous associations reported for pathological CSF A β 1-42 (PC ae C44:6, PC ae C44:5, and PC ae C44:4) in ROS/MAP at P-values significant after Bonferroni correction ($P < 2.94 \times 10^{-3}$) and with the same effect directions as in ADNI, despite the different measure for A β pathology. For 8 of the 14 sex- and APOE ϵ 4 status-stratified associations for A β and tau pathology, we observed non-zero effect heterogeneity estimates (12 of 1.4% - 45.7%) (Supplementary Table 4), albeit non-significant. The three metabolite measures in AIBL all showed non-zero effect heterogeneity estimates (12 of 39.7% - 54.3%) in the sex-stratified analyses with CSF p-tau (Supplementary Table 5), with effect heterogeneity being significant for SM (OH) C16:1 ($P = 0.016$). Combined, AIBL and ROS/MAP yielded non-zero heterogeneity

estimates for 2 out of 4 reported group comparisons for A β pathology and 8 out of 10 reported group comparisons for CSF p-tau and brain tau pathology.”

Given these additional evidences, we believe to provide the necessary context to evaluate the findings reported in our manuscript.

- iii) It's not clear how many analytes are testing. In the manuscript the authors mention In section 2.2 “Metabolite data acquisition” 186 metabolites. Then in section 2.3 “Metabolomics data processing”, they reported 182 metabolites. In addition, the description of the panel has a higher number of metabolites.

The reviewer is correct that, according to the manufacturer, the most up-to-date protocol in conjunction with specific instruments provides measurements of up to 188 metabolites. However, as established at the Duke Metabolomics Core, where the presented measurements have been obtained, the platform can yield valid measurements for up to 186 metabolites (as we stated in the original manuscript). Four of those metabolites have been removed due to technical issues during measurement prior to any downstream analysis. All of the remaining 182 analytes were subjected to several quality filters (% missing data, intra-class correlation, coefficient of variation), for which we now provide the determined QC measures in Supplementary Table 6. The number of metabolites included in the presented analyses after QC was given in section 2.3 (*Metabolomics data processing*), which, in the initial submission, concluded as follows: *“The final QC-ed metabolomics dataset was further restricted to individuals with data on all significant covariates (see Section 2.4. Phenotype data and covariate selection), resulting in the study dataset of 140 metabolites and 1,517 individuals.”*

Due to concern iv) raised by this reviewer, the underlined part changed to *“139 metabolites and 1,517 individuals”* in the revised manuscript. This number is also referenced in several instances across the manuscript and the supplement.

- iv) The authors are presenting data of analytes that were imputed for up to 40% of the subjects, which seems a too high percentage of missing values.

We take the point of the reviewer that this threshold may be considered as too high. In the revised manuscript, we adjusted this threshold to 20%, which leads to the exclusion of one additional metabolite (C4:1). This metabolite was not found to be significantly different between females and males, nor associated with/showing effect heterogeneity for the included biomarkers of AD. The results described in the initial manuscript are thus not affected by this adjustment. As mentioned before, to be fully transparent on the quality control process, we provide QC statistics for all metabolites in Supplementary Table 6. We further updated plots in Figure 1 to ascertain correct visualization of our findings.

- v) Do metabolites follow a normal distribution?

We agree that this is an important point for the validity of our results using regression/heterogeneity analysis. We specifically applied log₂-transformation to make sure that metabolite levels approximately follow a normal distribution. With the revised manuscript, we provide histograms of the distributions in Supplementary Figure 6 to be fully transparent about this.

- vi) Power is mentioned as a potential reason for lack of significance for expected associations, but no power analyses/simulation is presented.

The reviewer is correct in that we missed to provide a description of power calculations in the text, although we did include power calculations for the stratified analyses of metabolic sex-differences in the initial submission. We addressed this and added a description to the revised manuscript. To enable the reader to contextualize the replication analyses in ROS/MAP and AIBL appropriately, we also added thorough post-hoc power analyses to provide estimated sample sizes that would have been required to replicate our findings with equal power as in ADNI. We also clearly stated that this analysis is dependent on the assumption that effect sizes seen in the ADNI are generalizable to the replication cohort. The respective section in the revised methods now reads:

“In each power analysis, we transformed covariate-adjusted effect sizes to sample size-weighted standardized effects (Cohen’s d). For metabolic sex differences, we calculated the power for two-sample t-tests to identify significant sexual dimorphisms for metabolites with the standardized effect sizes observed in the pooled ADNI samples at Bonferroni significance in CN participants, participants with MCI, and patients with probable AD. To obtain estimates of sample sizes required to replicate metabolite associations and heterogeneity estimates, we used the same approach with power fixed to 50% (the post-hoc/observed power to find results at P-values equal or below the respective applied threshold, i.e. nominal or Bonferroni significance). Thereby, we estimated sample sizes assuming perfectly balanced datasets (with respect to sex and APOE ε4 status). This is a very rough approximation as it further assumes that the effect sizes reported for ADNI are generalizable to any replication cohort. Therefore, reported required sample sizes may deviate in reality.”

- vii) My understating is that the authors are comparing effects on males and females in Table 2, but some of the effects that are being compared are not significantly associated, indicating that effect can even change of direction.

As we describe in section 3.2 (*Sex-stratified analyses reveal substantial differences in the association of AD biomarkers and blood metabolite concentrations between men and women*), we are especially interested in metabolites that show significant heterogeneity of effects between sexes for a biomarker in question. To this end, we investigated the heterogeneity of all metabolite-biomarker associations that showed at least nominal significance ($P < 0.05$) in one stratum. The respective excerpt from section 3.2 reads: *“(iii) associations that showed suggestive significance ($P < 0.05$) in one sex coupled with significance for effect heterogeneity between female and male effect estimates.”*

We agree with the reviewer that, as some of the effects are not even nominally significant in one sex (or APOE ε4 status group), they may change effect direction. We do not draw conclusions concerning the means of the estimated effects if they are insignificant, but in contrast point to the fact that the estimated effect distributions differ significantly between males and females (and APOE ε4 status groups, respectively). These findings refer to a different kind of statistic and remain valid, independent of effect directions.

As this is a central concept of our study, we added a supplementary figure (Supplementary Figure 3) that is intended to clarify the basis of these analyses. We further removed any reference to effect directions from the manuscript and additionally tried to sharpen the definition and consistency of use of effect categories in both Figure 1 and the main text, which now reads:

“Based on this comparison, we classified metabolite–A-T-N biomarker associations into homogeneous effects if metabolites showed very similar effects in their association to the biomarker for both sexes (i.e., estimated heterogeneity p-value > 0.05), and heterogeneous effects if metabolites showed different effects in both sexes leading to significant heterogeneity (i.e., estimated heterogeneity p-value < 0.05) and/or sex-metabolite interaction (for more details on the concept of effect heterogeneity, see Supplementary Figure 3). Effects that were Bonferroni-significant in only one sex with either significant effect heterogeneity between males and females or significant sex-metabolite interaction were considered sex-specific.”

viii) Some of the results are already presented in Varma et al (2018) Plos Med.

While the reviewer is correct in that a subset of the data described in this work (p180 data in ADNI-1 participants) has been used as replication set in Varma et al., the present study has a very different focus that has been addressed in neither of our previous publications (or – to the best of our knowledge – any previous publication):

Varma et al. used brain metabolite concentrations to build classifiers distinguishing between pathology-free controls and AD patients. To evaluate their findings in the blood, they used the metabolites selected by the classifier (i.e., those that achieved the best predictive performance) and investigated their associations with AD or AD biomarkers in blood.

Here, we use data on two times more subjects than available for replication in Varma et al. by expanding to ADNI-GO/2 participants. More importantly, we specifically address the question of the dependency of metabolite associations on strong risk factors of AD. By that, our study provides an evaluation of differences in metabolite-biomarker associations between specific risk groups, thus identifying potential subgroup-specific effects/pathways linked to AD biomarker profiles. The presented work is therefore very different in both the applied approaches and the nature of the underlying research question.

The only conclusion we draw from findings that have been previously described is that we expect homogeneous associations to likely be consistent across different cohorts – and for this conclusion we provide independent replication (that has not been previously published).

Reviewer #3 (Remarks to the Author):

The major claims of the paper are the use of stratified association analyses on the metabolic aspects of AD, differencing between males and females and between APOE ε4 carriers and non-carriers. The results are novel and they would be of interest to other researchers in the AD field.

Some minor comments that I would like to be explained in detail are:

1. The use of targeted metabolomics kit with UPLC/MS/MS, instead of an untargeted analysis.

We thank the reviewer for this comment. Both targeted and non-targeted metabolomics studies have their advantages. Here, the use of the targeted p180 platform was chosen for the following reasons:

- A) **The p180 was used in several previous studies in AD and has been shown to be informative. Yet, comparison of several of the studies reveals inconsistencies in the reported associations. In order to study potential confounding effects that may have caused these observations, the p180 was selected. While we are aware that there are non-targeted platforms available with overlapping coverage, we considered the application of the same targeted platform as a more precise approach here – more so, as the p180 has been shown to yield very stable measurements across laboratories and instruments in ring trials.**
- B) **The number of measured metabolites is usually much higher when using non-targeted platforms. We are of course aware of the benefits of increasing the coverage in the human metabolome. However, the nature of this study already severely limits the power for detecting significant outcomes by conducting stratified analyses. Adding additional metabolites to the picture would have further increased the multiple testing burden.**

To note this aspect in our manuscript, we added the rather low number of included metabolites to the paragraph on limitations of our study:

“Finally, we only investigated two AD risk factors using a limited metabolomics panel focused on lipid and amino acid metabolism. Further risk factors, such as type 2 diabetes, cardiovascular disease, and high blood pressure, are linked to metabolic aspects as well and may reveal even greater molecular heterogeneity, in particular when expanding to additional metabolic pathways, e.g. by using non-targeted metabolomics.”

2. The removed plate batch effects using cross-plate mean normalization using NIST metabolite concentrations.

We added details for this procedure and an example visualization pre-/post-normalization to Supplementary Figure 5. The concept is to calculate dilution/correction factors for each plate and metabolite taking the overall mean of QC samples as reference and dividing by the per-plate mean of QC samples. The study sample measurements are then multiplied by the derived correction factors. The rationale behind using NIST samples for calculating the correction factors is to sustain real biological variance (that may e.g. be due to imperfect blinded sample randomization and that may be hiding in between-plate differences) while accounting for technical confounding. To ascertain that we were able to remove significant technical batch effects, we performed additional QC steps (filters for CV and ICC) after batch normalization.

3. The different types of study subjects, specifically MCI, are not indicated in the methods section.

We adjusted the respective part in the methods section to provide more details on the definition of diagnostic groups in ADNI:

“AD dementia diagnosis was established based on the NINDS-ADRDA criteria for probable AD for individuals with Mini-Mental State Exam (MMSE) scores between 20-26 (inclusive) and a Clinical Dementia Rating Scale (CDR) of 0.5 or 1.0. MCI participants did not meet these AD criteria and had MMSE scores between 24-30 (inclusive), a memory complaint, objective memory loss measured by education-adjusted scores on Wechsler Memory Scale Logical Memory II, a CDR of 0.5, absence of significant levels of impairment in other cognitive domains, and essentially preserved activities of daily living, meeting predetermined criteria for amnesic MCI.”

Reviewers' comments:

Reviewer #1 (Remarks to the Author):

No further questions from this reviewer.

Reviewer #2 (Remarks to the Author):

I much appreciate the authors effort in addressing very carefully my comments and critiques from the previous submission. Still I do have these concerns that I consider the authors should address:

1) The authors are now analyzing metabolites with a call rate higher than 20%, which is a much more stringent cutoff from the previous threshold (40%), and imputing the values for those subjects with no calls. I would suggest to investigate and report whether the effects are consistent, regardless of the significance, when only the subjects with effective measurement are analyzed, and thus discard the imputation as a possible source of bias. Is the percentage of missing rate associated with sex or APOE genotypes for the metabolites reported associated with ATN?

2) The similar effects reported for many of the metabolites reported in Tables 2 and 3 might indicate a high correlation level among the analytes associated to the distinct biomarkers. Can the authors perform a correlation analyses to illustrate how independent are the metabolites? How is the structure of the data? Are all of these analytes associated with ATN clustered in a single bin? Which are the pathways ascertained by the p188 that do not show significance association?

3) I would suggest the authors to indicate that the metabolites ascertained are in serum. For example, the first phrase of the results section would might be interpreted that the metabolites are from CSF and not serum

4) Are the effects reported also observed in ATN negative strata? This can help constrain the effects of the lifestyle has in the metabolite levels.

5) I would encourage authors to perform a meta-analyses to aggregate the evidence provided from each of the cohorts to obtain more precise insights of pathways associated.

Response to Reviewer Comments

We appreciate the continuing effort of the reviewer to help us improving our manuscript. In the below, we provide a point-by-point response to the comments.

I much appreciate the authors effort in addressing very carefully my comments and critiques from the previous submission. Still I do have these concerns that I consider the authors should address:

- 1) The authors are now analyzing metabolites with a call rate higher than 20%, which is a much more stringent cutoff from the previous threshold (40%), and imputing the values for those subjects with no calls. I would suggest to investigate and report whether the effects are consistent, regardless of the significance, when only the subjects with effective measurement are analyzed, and thus discard the imputation as a possible source of bias. Is the percentage of missing rate associated with sex or APOE genotypes for the metabolites reported associated with ATN?

We investigated the influence of imputation, although a threshold of maximal allowed missingness of 20% is, as the reviewer mentions, very strict when working with metabolomics data (here, it translates to 0.083% missing data points):

- **None of the effects were strongly influenced or driven by missing value imputation. See Table 1 attached to this document.**
 - **Neither sex ($P = 0.096$) nor APOE $\epsilon 4$ status ($P = 0.151$) was associated with percent missingness of measurements for the reported metabolites. We here included run day-dependent kit performance.**
- 2) The similar effects reported for many of the metabolites reported in Tables 2 and 3 might indicate a high correlation level among the analytes associated to the distinct biomarkers. Can the authors perform a correlation analyses to illustrate how independent are the metabolites? How is the structure of the data? Are all of these analytes associated with ATN clustered in a single bin? Which are the pathways ascertained by the p188 that do not show significance association?

There is a strong correlation structure observed for metabolites covered by the p180 kit. This is known and was reported several times, including in our previous work (Toledo et al., *Alzheimers Dement.* 2017 Sep;13(9):965-984. doi: 10.1016/j.jalz.2017.01.020). The pathways covered by the kit are very well described, including on the manufacturer's website and e.g. in the Toledo et al. paper cited above. For convenience, we attach a correlation heatmap to this document (Figure 1) that shows metabolites and the pathways for the data used in this study. In the present manuscript, we report and discuss associations with acylcarnitines, amino acids, phosphatidylcholines and sphingomyelins, which are covering most of the pathways of the p180 (exception are biogenic amines). Metabolites within one pathway generally show much stronger correlations than metabolites from distinct pathways, i.e. the reported analytes do not all lie "in a single bin". As the attached correlation heatmap is almost identical to the one reported in Figure 1 in Toledo et al. and the correlation structure of the kit replicates in healthy cohorts as well, we did not include this analysis/figure in the manuscript.

- 3) I would suggest the authors to indicate that the metabolites ascertained are in serum. For example, the first phrase of the results section would might be interpreted that the metabolites are from CSF and not serum.

We acknowledge that this is an important piece of information, which we consequently mention in many instances throughout the manuscript, including the abstract, as well as prominently in one of the result section headings (as requested by this reviewer before). We now added “serum” to the first phrase of the results section to avoid any misinterpretation.

- 4) Are the effects reported also observed in ATN negative strata? This can help constrain the effects of the lifestyle has in the metabolite levels.

By investigating continuous CSF p-tau levels and FDG-PET measurements across diagnostic groups, we intentionally ignored diagnostic assignments (which are naturally extremely correlated with biomarker positivity). As Table 1 in the manuscript shows, for CSF A β 1-42 there is also a quite substantial fraction of participants in ADNI showing pathological values that do not have a diagnosis of LMCI or AD (i.e., are not severely cognitively impaired).

Nevertheless, our dataset has almost equal sample sizes for participants with (n = 795) and without (n = 736) dementia or LMCI. If effects were solely driven by biomarker positivity leading to severe cognitive impairment and, hence, diagnosis of impairment associated with lifestyle changes, these effects would be significantly weakened by the less or non-impaired participants. This would, however, mean that we would see highly significant associations in a case-control study. This is something we looked into while curating the data for this manuscript. Yet, neither in the analysis of AD patients vs. cognitively normal (CN) controls, nor while looking at participants with LMCI vs. CN controls did we see any association surpassing the reported Bonferroni-adjusted significance threshold in any of the investigated stratifications for any of the reported metabolites.

- 5) I would encourage authors to perform a meta-analysis to aggregate the evidence provided from each of the cohorts to obtain more precise insights of pathways associated.

As we mentioned in the manuscript and in the first response to the reviewers, despite enormous efforts we were unable to find any cohort that featured the same AD biomarker readouts and simultaneously data from the same metabolomics kit as the ADNI. Prerequisites for a meta-analysis are streamlined data collection, processing and analysis protocols for variables included in the association studies in each cohort to ascertain valid outcomes. This is unfortunately impossible with the presented datasets, prohibiting us to follow this otherwise valuable suggestion.

Figure 1: Correlation heatmap of p180 metabolites.

REVIEWERS' COMMENTS:

Reviewer #2 (Remarks to the Author):

The authors have addresses all of my concerns